# FEATURE RESPONSIVENESS SCORES: MODEL-AGNOSTIC EXPLANATIONS FOR RECOURSE

**Seung Hyun Cheon**
UC San Diego

**Anneke Wernerfelt**
Haverford College

**Sorelle A. Friedler**
Haverford College

**Berk Ustun**
UC San Diego

## ABSTRACT

Machine learning models routinely automate decisions in applications like lending and hiring. In such settings, consumer protection rules require companies that deploy models to explain predictions to decision subjects. These rules are motivated, in part, by the belief that explanations can promote *recourse* by revealing information that individuals can use to contest or improve their outcomes. In practice, many companies comply with these rules by providing individuals with a list of the most important features for their prediction, which they identify based on feature importance scores from feature attribution methods such as SHAP or LIME. In this work, we show how these practices can undermine consumers by highlighting features that would not lead to an improved outcome and by explaining predictions that cannot be changed. We propose to address these issues by highlighting features based on their *responsiveness score* – i.e., the probability that an individual can attain a target prediction by changing a specific feature. We develop efficient methods to compute responsiveness scores for any model and any dataset. We conduct an extensive empirical study on the responsiveness of explanations in lending. Our results show that standard practices in consumer finance can backfire by presenting consumers with *reasons without recourse*, and demonstrate how our approach improves consumer protection by highlighting responsive features and identifying fixed predictions.

## 1 INTRODUCTION

Machine learning models are routinely used to automate and support decisions about people in consumer finance [30], employment [9, 47], and public services [64, 19, 27]. In these domains, companies are increasingly expected – and in some cases legally required – to provide explanations to individuals affected by their predictions [1, 61, 55, 20]. In the United States, for example, the *adverse action* provision in the Equal Credit Opportunity Act mandates that lenders provide a list of "principal reasons" to consumers who are denied credit [1]. In the European Union, Article 86 of the AI Act [20] grants individuals a *right to explanation* in "high risk" domains [see Annex III of 20].

Explanations are a cornerstone of consumer protection in such settings because they may reveal information that consumers can use to exercise broader rights [16]. In the United States, for example, adverse action notices are meant to support: *anti-discrimination* by revealing that a prediction was influenced by protected characteristics; *rectification*, by revealing that a prediction was based on erroneous information; and *recourse*, by revealing how they could attain a desired prediction in the future [54, 50]. In the European Union, the right to an explanation in the GDPR is meant to reveal information that consumers could use to contest their decisions or request human review [33].

Many explainability mandates are developed in the absence of clear directives regarding the appropriate structure and construction methodology for explanations. This ambiguity exists partly because many mandates are new – with policy makers debating how to enforce them – or have not yet become law. In the United States, where the adverse action requirement has been in place for over half a century, companies provide consumers with feature-highlighting explanations that list the most important features for their prediction. They typically generate these explanations using feature attribution methods such as SHAP [40] and LIME [48], scoring features in terms of their relative contribution to the prediction and including the top-scoring features in a letter to consumers. This approach has been recognized as a principled attempt to identify "principal reasons" and has gained wide-spread adoption due to its simplicity [see e.g., recommendations in 24].

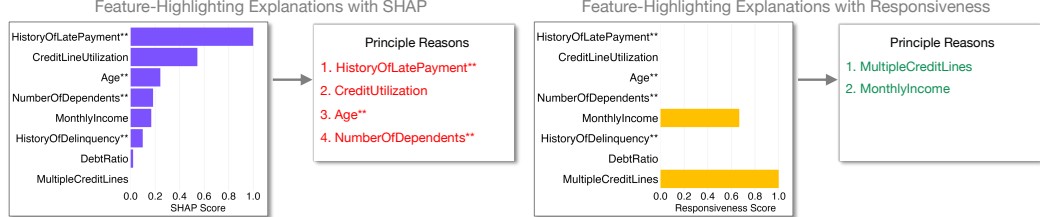

**Figure 1:** Feature-highlighting explanations for a person denied credit by an XGBoost model on the `givemecredit` dataset (see Section 4). We construct each explanation by highlighting up to four features with the largest SHAP scores (left) and responsiveness scores (right). As shown, an explanation built with SHAP highlights four features that are immutable** (e.g., `Age`) or unresponsive – no intervention leads to a target prediction (e.g., `CreditUtilization`). In contrast, an explanation built with responsiveness scores returns the *only* 2 features that can be changed to attain a desired prediction: `MonthlyIncome` and `MultipleCreditLines`. The score for `MultipleCreditLines` is 1, which means any intervention would lead to a target prediction.

In this work, we analyze how explanations can effectively achieve one of their goals: helping consumers attain recourse. Our work is motivated by the fact that explainability mandates are designed to achieve multiple goals and yet overlook implementation details. To this end, we assess the efficacy of feature attribution methods in supporting recourse, and develop a targeted approach for this objective. Our main contributions include:

1. We identify an inherent limitation of feature attribution methods in consumer-facing applications – *reasons without recourse* – where we present features in explanations, yet modifying them yields no path to the desired outcome.

2. We introduce a new approach to explain individual predictions by measuring feature responsiveness – the probability of attaining recourse through a randomly chosen intervention on its features.

3. We conduct an empirical study on feature-highlighting explanations in consumer finance, finding that standard methods can harm consumers by highlighting immutable and unresponsive features. Our framework improves upon feature attribution methods, promoting recourse and transparency by highlighting features that help attain target predictions and flagging predictions that are difficult or impossible to change. Our analysis also reveals the need for additional information in explanations to effectively facilitate recourse.

4. We include a Python library to compute feature responsiveness scores available on GitHub.

**Related Work**  Our work is broadly related to a stream of methods to explain individual predictions [see, e.g., 48, 40, 41, 36]. We identify these methods can inflict harm in consumer-facing applications by providing individuals with *reasons without recourse*. We view reasons without recourse as a structural limitation that affects how we operationalize explainability mandates, akin to limitations of explainability that arise due to the multiplicity of predictions [42, 60, 8], the indeterminacy of explanations [11, 39], and the potential for fairwashing [4, 52, 28].

Our goal is broadly inspired by work in algorithmic recourse, as we seek to highlight how individuals can change their predictions [57, 34, 58]. Existing methods in this area are designed to return an action that an individual could perform to attain a target prediction. In contrast, our method estimates the proportion of actions on each feature that lead to a target prediction (see Fig. 2). We construct these estimates by sampling or enumerating a set of reachable points [38] and can be adapted to address practical challenges related to causality [35, 14, 26] and distributional robustness [45, 46, 56].

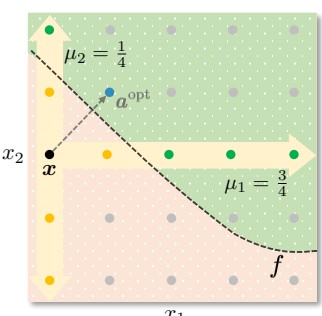

**Figure 2:** Standard methods for recourse provision return the closest action that would attain the target prediction $a^{\text{opt}}$. Our method estimates the proportion of actions on each feature that attain the target prediction. Here, $\mu_1 = \frac{3}{4}$ and $\mu_2 = \frac{1}{4}$ because $x$ could attain the target prediction through 3/4 possible actions on $x_1$ and 1/4 possible actions on $x_2$.

## 2 PROBLEM STATEMENT

In what follows, we discuss how companies comply with rules and regulations that require them to explain the predictions of a model to individuals. We consider a classification task where a company uses a model $f : \mathcal{X} \to \mathcal{Y}$ to predict a label $y \in \mathcal{Y}$ from a set of $d$ *features* $\boldsymbol{x} = [x_1, x_2, \ldots, x_d] \in \mathcal{X} \subseteq \mathbb{R}^d$. We restrict our attention to tasks where each instance represents a person, and where their features encode characteristics that are semantically meaningful for the task at hand. In such tasks, we can safely assume that features are bounded.[1]

We define the scope of explainability mandates in terms of a *target prediction* $\hat{y}^{\text{target}} \in \mathcal{Y}$. We assume that the target prediction $\hat{y}^{\text{target}}$ represents a desirable outcome, e.g., $f(\boldsymbol{x}) = \hat{y}^{\text{target}} = 1$ if a person with features $\boldsymbol{x}$ will repay their loan, and that any other prediction represents an adverse outcome. Under these conventions, companies must provide explanations to all individuals with features $\boldsymbol{x}$ such that $f(\boldsymbol{x}) \neq \hat{y}^{\text{target}}$. Informally, these practices would support *recourse* [57] if they include information that each person could use to attain a target prediction – i.e., to update their features to a point $\boldsymbol{x}'$ such that $f(\boldsymbol{x}') = \hat{y}^{\text{target}}$.

| Feature Values | | Label Counts | | Best Predictions |
|---|---|---|---|---|
| age $\geq 60$ | savings $\geq$ 60K | $n_0$ | $n_1$ | $f(x_1, x_2)$ |
| 0 | 0 | 40 | 10 | 0 |
| 0 | 1 | 10 | 30 | 1 |
| 1 | 0 | 20 | 10 | 0 |
| 1 | 1 | 30 | 10 | 0 |

**Table 1:** Stylized classification task where the most accurate model assigns *fixed predictions* due to the immutable feature age $\geq 60$. We train a model to predict $y =$ repayment from binary features $(x_1, x_2) = ($age $\geq 60$, savings $\geq$ 60K$)$ on a dataset with $n_0$ negative labels and $n_1$ positive labels. Here, individuals with age $\geq 60 = 1$ can change their feature values, but cannot change their prediction as the model outputs $f(x_1, x_2) = 0$ for $(x_1, x_2) \in \{(1, 0), (1, 1)\}$.

**Feature-Highlighting Explanations**   Companies comply with explainability mandates by constructing a *feature-highlighting explanation* – i.e., a list that contains the most important features that contribute to a prediction [5]. In practice, companies select which features to highlight based on feature importance scores that they obtain from a feature attribution method [24, 22]:

**Definition 1.** Given a model $f : \mathcal{X} \to \mathcal{Y}$ and a point $\boldsymbol{x} \in \mathcal{X}$, a *feature attribution method* is a function $\boldsymbol{\phi} : \mathcal{X} \to \mathbb{R}^d$ that returns a vector of *feature importance scores* $\boldsymbol{\phi}(\boldsymbol{x}; f) := [\phi_1(\boldsymbol{x}; f), \ldots, \phi_d(\boldsymbol{x}; f)]$ where $\phi_j(\boldsymbol{x}; f)$ reflects the relative contribution of feature $j$ towards the prediction $f(\boldsymbol{x})$. We write $\boldsymbol{\phi}(\boldsymbol{x})$ instead of $\boldsymbol{\phi}(\boldsymbol{x}; f)$ when $f$ is clear from context.

We can use the function $\boldsymbol{\phi} : \mathcal{X} \to \mathbb{R}^d$ to represent major classes of local explainability methods:

- *Local Surrogate Methods* [48, 66, 65, 17, 53], which explain the prediction of a model $f$ at a point $\boldsymbol{x}$ by fitting an surrogate model to approximate the decision boundary of $f$ near $\boldsymbol{x}$. Given the surrogate model, we can use its parameters to determine the importance of each feature $\phi_j(\boldsymbol{x})$.

- *Shapley Value Methods* [see e.g., 40, 31, 25], which cast the features of a model $f$ as players in a cooperative game, and estimate $\phi_j(\boldsymbol{x})$ as the marginal contribution of feature $j$ towards the prediction $f(\boldsymbol{x})$ under axioms of social choice [51].

Scores from these models indicate relative importance due to the following properties:

- *Relevance*: A feature with an attribution score $\phi_j(\boldsymbol{x}) = 0$ can be changed arbitrarily without changing the prediction for $\boldsymbol{x}$ [see e.g., the "missingness" axiom in 40].

- *Strength*: Given two features $j, k \in [d]$ such that $|\phi_j(\boldsymbol{x})| > |\phi_k(\boldsymbol{x})|$, feature $j$ has a stronger contribution to the prediction than feature $k$ [see e.g., 44].

Given top-scoring features, companies can automatically convert them into natural language explanations for decision subjects [e.g., a reason code 21, 13].

**Reasons without Recourse**   Feature attribution methods can highlight features that genuinely reflect the "principal reasons" or "main factors" for each prediction. In practice, however, these methods may fail to highlight features that lead to a desired outcome:

- *Inability to Characterize Counterfactual Behavior*: Feature attribution methods often assign high importance scores to features that are not indicative of the model's local behavior. Bilodeau et al.

---

[1]In practice, many features will be bounded by definition – e.g., a binary feature such as recent_payment $\in \{0, 1\}$. In other cases, we can set loose bounds that apply to all decision subjects – e.g., age $\in [0, 120]$ or income $\in [0, 10^9]$.

| Requirement | Example | Features | Actionability Constraint |
|---|---|---|---|
| Immutability | `age` cannot change | $x_j = $ `age` | $v_j = 0$ |
| Monotonicity | `recent_payment` can only increase | $x_j = $ `recent_payment` | $v_j \geq 0$ |
| Integrality | `late_payments` must be positive integer $\leq 12$ | $x_j = $ `late_payments` | $v_j \in \mathbb{Z}^+ \cap [0 - x_j, 12 - x_j]$ |
| Encoding Validity | preserve one-hot encoding of categorical feature `housing` $\in \{$`own`, `rent`, `other`$\}$ | $x_k = \mathbb{1}[\text{housing=own}]$ $x_l = \mathbb{1}[\text{housing=rent}]$ $x_m = \mathbb{1}[\text{housing=other}]$ | $v_j + x_j \in \{0, 1\}$ for $j \in \{k, l, m\}$ $\sum_{j \in \{k,l,m\}} v_j + x_j = 1$ |
| Logical Implication | if `has_savings_account` $=$ `TRUE` then `savings_balance` $\geq 0$ else `savings_balance` $= 0$ | $x_j = $ `has_savings_account` $x_k = $ `savings_balance` | $v_j + x_j \in \{0, 1\}$ $v_k + x_k \in [0, 10^{12}]$ $v_k + x_k \leq 10^{12}(x_j + v_j)$ |
| Causal Implication | if `years_of_account_history` increases then `age` will increase commensurately | $x_j = $ `years_of_account_history` $x_k = $ `age` | $x_j + v_j \leq x_k + \delta_k$ $\delta_k \in [0, 100]$ |

**Table 2:** Examples of actionability constraints on semantically meaningful features for a lending task. Each constraint can be expressed in natural language and embedded into an optimization problem using standard techniques in mathematical programming [see, e.g., 62]. See Appendix B for more examples.

[7] demonstrate that methods like SHAP suffer from under-specification – i.e., a feature with a given level of responsiveness can receive different attribution scores – and are sensitive to baseline values, which may not be relevant in describing model behavior locally.

- *Ignorance of Actionability*: Feature attribution methods do not account for how individuals can change their features. This can lead them to highlight features that are immutable or even highlight features when individuals are assigned fixed predictions – $f(\boldsymbol{x}') \neq \hat{y}^{\text{target}} \; \forall \, \boldsymbol{x}'$ (see Table 1).

Highlighting important but unresponsive features undermines the value of explainability mandates. For example, feature attribution methods might highlight immutable features, even when recourse is available through other features. Alternatively, they may highlight features that seem actionable but do not lead to the desired prediction, causing individuals to focus on ineffective changes.

## 3 Measuring Feature Responsiveness

Our goal is to measure the responsiveness of the model with respect to independent changes in each feature. First, we characterize how each feature can be changed:

**Characterizing Actionability** Let $\boldsymbol{x}$ be the feature vector representing a person with the capability of changing their associated data to new point $\boldsymbol{x}'$:

$$\boldsymbol{x}' = \boldsymbol{x} + \boldsymbol{v} + \boldsymbol{z}(\boldsymbol{v}) \tag{1}$$

where $\boldsymbol{v} \in \mathbb{R}^d$ is an intervention with one non-zero element $v_j$, and $\boldsymbol{z}(\boldsymbol{v})$ is the *downstream effect* of $\boldsymbol{v}$ that captures how other features change as a result of the intervention $\boldsymbol{v}$. Given a model that includes the features like `yrs_account_history` and `age`, an intervention that increases `yrs_account_history` should lead to commensurate increase in `age` – a downstream effect.

Each feature is *inherently constrained* by physical limits (i.e., bounds), type (e.g., one-hot encoding), or nature (e.g., monotonicity of ordinal encoding of `education_level`). As shown in Table 2, we can elicit these constraints from human experts in natural language, and convert them into equations that we can embed into optimization problems [e.g., to search for recourse actions 57, 38]. In fact, we can extend beyond inherent constraints and encode non-trivial constraints like the causal relationship between `yrs_account_history` and `age` mentioned earlier. We refer to the set of such constraints, both inherent and non-trivial, as *actionability constraints*.

We describe the set of points we can reach by intervening on a feature $j$ within these constraints:

**Definition 2.** Given a point $\boldsymbol{x}$, we define the set of *interventions* on feature $j \in [d]$:

$$V_j(\boldsymbol{x}) := \left\{\boldsymbol{v} \in \mathbb{R}^d \,|\, v_j \neq 0, v_j \text{ is feasible under the actionability constraints}, v_k = 0 \text{ for } k \neq j \right\}$$

We also define the *action set*, which describes the set of points we can reach by intervening on $j$:

$$A_j(\boldsymbol{x}) := \{\boldsymbol{a} = \boldsymbol{v} + \boldsymbol{z}(\boldsymbol{v}) \mid \boldsymbol{v} \in V_j(\boldsymbol{x})\}$$

We are now ready to introduce our main technical contribution – the *responsiveness score*.

| Reachable Points | | | Prediction | | Responsiveness Score |
|---|---|---|---|---|---|
| n_loans | guarantor | age | repay | | |
| $x_1$ | $x_2$ | $x_3$ | $f(\boldsymbol{x})$ | | |
| 3 | 0 | 24 | 0 | $\boldsymbol{x}$ | $\mu_1(\boldsymbol{x}) = \dfrac{1}{\lvert R_1(\boldsymbol{x})\rvert}\displaystyle\sum_{\boldsymbol{x}'\in R_1(\boldsymbol{x})}\mathbb{1}[f(\boldsymbol{x}')=1] = \dfrac{2}{3}$ |
| 2 | 0 | 24 | 1 | | |
| 1 | 0 | 24 | 0 | $R_1(\boldsymbol{x})$ | $\mu_2(\boldsymbol{x}) = \dfrac{1}{\lvert R_2(\boldsymbol{x})\rvert}\displaystyle\sum_{\boldsymbol{x}'\in R_2(\boldsymbol{x})}\mathbb{1}[f(\boldsymbol{x}')=1] = 1$ |
| 0 | 0 | 24 | 1 | | |
| 3 | 1 | 24 | 1 | $R_2(\boldsymbol{x})$ | |
| 2 | 1 | 24 | 1 | | $\mu_3(\boldsymbol{x}) = 0$ |
| 1 | 1 | 24 | 1 | $R_3(\boldsymbol{x}) = \varnothing$ | |
| 0 | 1 | 24 | 1 | | |

**Figure 3:** Responsiveness scores for a stylized task with three features (n_loans, guarantor, age). We show all points that can be reached for a person with features $\boldsymbol{x} = (3, 0, 24)$. Here, the reachable set $R_j(\boldsymbol{x})$ for each of the feature contains the set of points that can be reached through single feature interventions, and $R_3(\boldsymbol{x}) = \varnothing$ because age is immutable. Given these sets, we compute the responsiveness scores for each feature by querying its predictions over points in their respective reachable set $R_j(\boldsymbol{x})$.

**Definition 3 (Responsiveness Score ($\mu_j(\boldsymbol{x})$)).** Given a model $f : \mathcal{X} \to \mathcal{Y}$ that assigns an adverse prediction to a point $\boldsymbol{x} \in \mathcal{X}$, we define the *responsiveness score* of feature $j$ as:

$$\mu_j(\boldsymbol{x} \mid f, A_j(\boldsymbol{x})) := \Pr(f(\boldsymbol{x} + \boldsymbol{a}) = \hat{y}^{\text{target}} \mid \boldsymbol{a} \in A_j(\boldsymbol{x}))$$

We write $\mu_j(\boldsymbol{x})$ instead of $\mu_j(\boldsymbol{x} \mid f, A_j(\boldsymbol{x}))$ when $f$ and $A_j(\boldsymbol{x})$ are clear from context.

Responsiveness scores reflect the proportion of single-feature actions on feature $j$ that result in $\hat{y}^{\text{target}}$. A score of $\mu_j(\boldsymbol{x}) = 0$ indicates that changing feature $j$ cannot achieve $\hat{y}^{\text{target}}$, while $\mu_j(\boldsymbol{x}) = 1$ means any intervention on it will. These values depend on actionability constraints, which, in the simplest case, encode indisputable constraints related to feature encoding or physical limits, meaning that $\mu_j(\boldsymbol{x})$ represents an upper bound on the true responsiveness of feature $j$.

$\mu_j(\boldsymbol{x})$ can be interpreted as the probability of a person $\boldsymbol{x}$ achieving $\hat{y}^{\text{target}}$ after a random intervention on feature $j$. Hence, feature-highlighting explanations using responsiveness scores is optimal because these explanations do not specify the necessary degree of change; highlighting the most responsive feature maximizes the likelihood of recourse. We only include features that, when changed independently, lead to recourse, enabling consumers to choose any highlighted feature.

This approach offers stronger protections by: (1) providing explanations *only* to individuals with recourse, and (2) detecting potentially misleading or uninformative feature-highlighting explanations.

**Remark 1.** *Given a model $f : \mathcal{X} \to \mathcal{Y}$, let $\mu_1(\boldsymbol{x}) \ldots \mu_d(\boldsymbol{x})$ denote the feature responsiveness scores for an individual $\boldsymbol{x} \in \mathcal{X}$. If $\mu_j(\boldsymbol{x}) = 0$ for all $j \in [d]$, then either: (a) $f$ assigns a fixed prediction to $\boldsymbol{x}$, or (b) $f$ can only provide recourse to $\boldsymbol{x}$ through a joint intervention on two or more features.*

According to Remark 1, $\mu_j(\boldsymbol{x}) = 0$ for all $j \in [d]$ in two scenarios. We can triage these cases to mitigate harm. For fixed predictions (case (a)), we can withhold explanations and notify developers or regulators. For predictions requiring joint interventions (case (b)), we can include a warning against assuming feature independence.

### 3.1 Computing Scores with Reachable Sets

The key challenge in computing responsiveness scores is to include actionability constraints in the calculation. As such, we present an alternative representation of actionability, a *reachable set*:

**Definition 4.** Given a point $\boldsymbol{x}$, the *reachable set* for feature $j$ contains all feature vectors that can be attained through actions on feature $j$: $R_j(\boldsymbol{x}) := \{\boldsymbol{x} + \boldsymbol{a} \mid \boldsymbol{a} \in A_j(\boldsymbol{x})\}$.

Given a reachable set $R_j(\boldsymbol{x})$, we can calculate the responsiveness score for *any* model by querying its predictions over reachable points (see Fig. 3) as

$$\mu_j(\boldsymbol{x} \mid f, A_j(\boldsymbol{x})) := \mathbb{E}_{\boldsymbol{x}' \sim R_j(\boldsymbol{x})}\big[\mathbb{1}[f(\boldsymbol{x}') = \hat{y}^{\text{target}}]\big] \tag{2}$$

where $\mathbb{1}$ is the indicator function. In such cases, we only need to compute the reachable sets *once* and use it to measure feature responsiveness for any model. This enables us to take responsiveness into account in various stages of model development (i.e., model selection). Here, we outline two methods to construct reachable sets and include additional details in Appendix A.1.

**Sampling** We present a procedure to sample reachable points in Algorithm 1. Given a point $\boldsymbol{x}$ and the action set $A_j(\boldsymbol{x})$, this procedure returns a uniform sample of $N$ reachable points by rejection sampling. In Line 2, it calls the SampleAction$(\boldsymbol{x}, A_j)$ routine to propose a candidate action vector $\hat{\boldsymbol{a}}$ that obeys separable actionability constraints such as bounds and integrality constraints. In Line 3, it calls the CheckFeasibility$(\boldsymbol{x}, \hat{\boldsymbol{a}}, A_j)$ routine to check if $\hat{\boldsymbol{a}}$ obeys joint actionability constraints by solving a mixed-integer program. The procedure terminates after it has found $N$ feasible actions $\hat{\boldsymbol{a}} \in A_j(\boldsymbol{x})$, each of which can define a reachable point as $\boldsymbol{x} + \hat{\boldsymbol{a}}$. Given $R_j(\boldsymbol{x})$, we can recover an unbiased estimate for the responsiveness score for feature $j$ for a model $f$ as $\hat{\mu}_j(\boldsymbol{x}) := \frac{1}{N} \sum_{\boldsymbol{x}' \in \hat{R}_{\boldsymbol{x}}(j)} \mathbb{1}[f(\boldsymbol{x}') = \hat{y}^{\text{target}}]$. We can set the sample size $N$ to control the reliability with which we flag predictions that satisfy the conditions in Remark 1, which warrant more detailed explanations (see Appendix A.2).

**Enumeration** We present an alternative procedure that can be used to enumerate a complete reachable set in Algorithm 2 to enumerate $R_j(\boldsymbol{x})$. This procedure can only be applied to construct a reachable set for features that are discrete and

---

**Algorithm 1** Sample Reachable Points

**Require:** $\boldsymbol{x} \in \mathcal{X}$            *point*
**Require:** $A_j(\boldsymbol{x})$       *action set for feature j*
**Require:** $N \in \mathbb{N}$     *sample size (see Appendix A.2)*
      $\hat{R}_j \leftarrow \varnothing$
1: **repeat**
2:     $\hat{\boldsymbol{a}} \leftarrow$ SampleAction$(\boldsymbol{x}, A_j(\boldsymbol{x}))$
3:     **if** CheckFeasibility$(\boldsymbol{x}, \hat{\boldsymbol{a}}, A_j(\boldsymbol{x}))$ **then**
4:        $\hat{R}_j \leftarrow \hat{R}_j \cup \{\boldsymbol{x} + \hat{\boldsymbol{a}}\}$
5:     **end if**
6: **until** $|\hat{R}_j| = N$
**Output** $\hat{R}_j$     *N reachable points via actions on j*

---

**Algorithm 2** Enumerate Reachable Points

**Require:** $\boldsymbol{x} \in \mathcal{X}$        *point*
**Require:** $A_j(\boldsymbol{x})$    *action set for feature j (discrete)*
**Require:** $\tilde{V}_j(\boldsymbol{x})$    *finite set of interventions for feature j*
      $R_j \leftarrow \varnothing$
      $A_j \leftarrow A_j(\boldsymbol{x})$
1: **for each** $\boldsymbol{v} \in \tilde{V}_j(\boldsymbol{x})$ **do**
2:     **repeat**
3:       $\boldsymbol{a}^* \leftarrow$ Find1DAction$(\boldsymbol{x}, \boldsymbol{v}, A_j)$
4:       $R_j \leftarrow R_j \cup \{\boldsymbol{x} + \boldsymbol{a}^*\}$
5:       $A_j \leftarrow A_j \setminus \{\boldsymbol{a}^*\}$
6:     **until** Find1DAction$(\boldsymbol{x}, \boldsymbol{v}, A_j)$ is infeasible
7: **end for**
**Output** $R_j$    *all reachable points via actions in j*

---

whose downstream features are also discrete. The algorithm enumerates reachable points for a feature $j$ by repeatedly solving the discrete optimization problem:

$$\text{Find1DAction}(\boldsymbol{x}, \boldsymbol{v}, A_j) := \operatorname*{argmin}_{\boldsymbol{a} \in A_j(\boldsymbol{x})} \|\boldsymbol{a}\|_1 \text{ s.t. } a_j = v_j$$

for each $\boldsymbol{v} \in \tilde{V}_j(\boldsymbol{x})$ (Line 1). $\tilde{V}_j(\boldsymbol{x})$ is a superset of $V_j(\boldsymbol{x})$ (i.e., $V_j(\boldsymbol{x}) \subseteq \tilde{V}_j(\boldsymbol{x})$), where it contains possibly infeasible interventions. Generally, we can deduce $\tilde{V}_j(\boldsymbol{x})$ from feature bounds and monotonicity constraints. We formulate Find1DAction$(\boldsymbol{x}, \boldsymbol{v}, A_j)$ as a mixed-integer program that we update at each iteration with a "no good constraint" to remove previous solutions (Line 5). We store each of the enumerated actions into $R_j(\boldsymbol{x})$, which can be used to calculate exact responsiveness scores. The procedure adapts a method used to enumerate a complete reachable set for all features in Kothari et al. [38], and is more tractable as we only enumerate single-feature interventions and their downstream effects.

**Extensions** One of the benefits of using reachable sets to compute responsiveness scores is that we easily customize scores to meet additional requirements (see e.g., Section 5). One such requirement is *monotonicity* – i.e., whether a person would be guaranteed a target prediction after increasing (or decreasing) a feature beyond threshold value. In general, we can account for such properties through simple operations such as filtering or weighing reachable points. We can also apply these operations to construct responsiveness scores that address practical challenges in settings where we are given additional inputs:

- *Individual Preferences*: Given a cost function that captures the difficulty of actions in each direction, we can highlight features that are easier to change (i.e., least costly $k$ features) using a cost-weighed score: $\mu_j^{\text{cost}}(\boldsymbol{x}; \text{ cost}) = \sum_{\boldsymbol{a} \in A_j(\boldsymbol{x})} \text{cost}(\boldsymbol{a}; \boldsymbol{x}) \cdot \mathbb{1}[f(\boldsymbol{x} + \boldsymbol{a}) = \hat{y}^{\text{target}}]$.

- *Distributional Robustness*: Given a general reachable set $R(\boldsymbol{x})$ that contains all points that we can reach, we can highlight features can be changed to attain a target prediction regardless of how other features change by computing a *robust score*: $\mu_j^{\text{robust}}(\boldsymbol{x}) = \min_{\boldsymbol{\delta} \in \Delta_{-j}} \Pr(f(\boldsymbol{x} + \boldsymbol{a} + \boldsymbol{\delta}) = \hat{y}^{\text{target}} \mid \boldsymbol{a} \in A_j(\boldsymbol{x}))$, Here, $\Delta_{-j} := \{\boldsymbol{\delta} \in \mathbb{R}^d \mid \delta_j = 0, \|\boldsymbol{\delta}\| < \varepsilon\}$ is the set of perturbations on features other than $j$ given a budget $\varepsilon > 0$ [see e.g., 45].

## 4 EXPERIMENTS

We present an empirical study on the responsiveness of explanations. Our goals are to demonstrate the limitations of existing feature attribution methods, and show how our approach can support recourse and flag fixed predictions. We include additional details and results in Appendix B, and code to reproduce our results on GitHub.

**Setup**  We work with three publicly available consumer finance classification datasets. Here, each instance represents a consumer and the label indicates if they will repay a loan. We consider discrete version of each dataset in which we can compute exact responsiveness scores and certify existence of recourse. Given these features, we define *inherent actionability constraints* that reflect indisputable requirements that apply to all individuals (e.g., that disallow changes to immutable attributes, preserve feature encodings, and adhere to deterministic causal effects).

We split each dataset into a training sample (80%; to train models and tune parameters) and a test sample (20%; to evaluate out-of-sample performance). We fit models using (1) *logistic regression* (LR), (2) *XGBoost* (XGB), and (3) *random forests* (RF). For each model, we construct feature-highlighting explanations for each person who is denied credit that highlight up to *four features*, which reflects the recommended number of reasons to include in adverse action notice by the U.S. Consumer Finance Protection Bureau [see 2]. We include the top-4 scoring features from the following methods:

- *Feature Responsiveness* (RESP): We compute responsive scores by enumerating reachable sets using Algorithm 2 with respect to inherent actionability constraints in Appendix B.
- *Standard Feature Attribution*: We consider model-agnostic methods that are widely used in the lending industry [24]: SHAP [40]; and LIME [48].
- *Actionable Feature Attribution*: We consider *action-aware* variants of feature attribution methods, SHAP-AW and LIME-AW which aim to highlight more responsive features by setting $\phi_j(\boldsymbol{x}) \leftarrow 0$ for features that are not actionable (i.e. can't be changed).

| Dataset | Metrics | LR | RF | XGB |
|---|---|---|---|---|
| heloc | % Denied | 56.1% | 58.3% | 57.0% |
| $n = 5,842$ | ↳ % No Recourse | 22.2% | 31.3% | 53.1% |
| $d = 43$ | ↳ % 1-D Recourse | 41.0% | 31.7% | 25.3% |
| FICO [23] | ↳ % $n$-D Recourse | 36.8% | 37.0% | 21.6% |
| german | % Denied | 22.9% | 17.5% | 22.0% |
| $n = 1,000$ | ↳ % No Recourse | 7.4% | 28.6% | 11.8% |
| $d = 36$ | ↳ % 1-D Recourse | 74.2% | 48.0% | 68.2% |
| Dua & Graff [15] | ↳ % $n$-D Recourse | 18.3% | 23.4% | 20.0% |
| givemecredit | % Denied | 24.6% | 24.7% | 24.8% |
| $n = 120,268$ | ↳ % No Recourse | 15.6% | 0.2% | 11.5% |
| $d = 23$ | ↳ % 1-D Recourse | 72.4% | 93.2% | 76.0% |
| Kaggle [32] | ↳ % $n$-D Recourse | 12.0% | 6.6% | 12.5% |

**Table 3:** Fraction of $n$ individuals who would receive an explanation as a result of an adverse prediction (% *Denied*) for each dataset and model. We characterize the potential to highlight features that lead to recourse by reporting the following metrics:% *Fixed*, % of denied individuals who with a fixed prediction (in red) – i.e., who have no recourse under any explanation; % *1-D*, % of denied individuals who can achieve recourse by intervening on a single feature – i.e., who could have recourse from feature-highlighting explanations; and % *n-D*, the fraction of denied individuals who can only achieve recourse by changing 2 or more features;

**On the Limits of Feature Highlighting Explanations**  Our results in Table 3 highlight how widespread practices to comply with explainability mandates can help consumers achieve recourse. For example, given an LR model on the heloc dataset, a lender would provide feature-highlighting explanations to 56.1% of individuals who are denied a loan. Among them: 41.0% can change a single feature to attain a desired prediction; 36.8% can only achieve recourse by changing 2 or more features simultaneously; and 22.2% are assigned a fixed prediction. Although the magnitude of these segments can vary considerably across datasets and model classes, there is no case where every person who is denied credit can change their prediction with a single-feature intervention. Some will require joint interventions. Others will have no path to recourse.

These results reflect *best-case estimates* of providing recourse with feature-highlighting explanations. Specifically, the 41.0% of individuals who could achieve recourse by a single feature intervention could only do so if we used an *ideal* method that assigns the highest scores to responsive features, and did not face additional actionability constraints. They also characterize where feature attribution methods may fail – i.e., 22.2% of denied individuals would be misled by any explanation as their prediction is fixed. These results underscore the need to impose stricter guidelines on compliance and devise an alternative notification framework for individuals fixed predictions.

**On Explanations with Feature Attribution Scores**  Our results show how standard methods for feature attribution can highlight features that are uninformative or misleading. Given the LR model

| | | LR | | | | | XGB | | | | |
| | | All Features | | Actionable Features | | | All Features | | Actionable Features | | |
| Dataset | Metrics | LIME | SHAP | LIME-AW | SHAP-AW | RESP | LIME | SHAP | LIME-AW | SHAP-AW | RESP |
|---|---|---|---|---|---|---|---|---|---|---|---|
| heloc | % Presented with Explanations | 100.0% | 100.0% | 100.0% | 100.0% | 41.0% | 100.0% | 100.0% | 100.0% | 100.0% | 25.3% |
| $n = 5,842$ | ↳ % All Features Unresponsive | 92.7% | 77.3% | 76.8% | 70.3% | 0.0% | 93.2% | 82.3% | 80.0% | 79.6% | 0.0% |
| $d = 43$ features | ↳ % At Least 1 Feature Responsive | 7.3% | 22.7% | 23.2% | 29.7% | 100.0% | 6.8% | 17.7% | 20.0% | 20.4% | 100.0% |
| $d_A = 31$ mutable | ↳ % All Features Responsive | 0.0% | 0.0% | 0.0% | 0.2% | **100.0%** | 0.0% | 0.0% | 0.0% | 0.0% | **100.0%** |
| FICO [23] | ↳ # Features Highlighted | 4.0 | 4.0 | 4.0 | 4.0 | 2.3 | 4.0 | 4.0 | 4.0 | 4.0 | 2.5 |
| german | % Presented with Explanations | 100.0% | 100.0% | 100.0% | 100.0% | 74.2% | 100.0% | 100.0% | 100.0% | 100.0% | 68.2% |
| $n = 1,000$ | ↳ % All Features Unresponsive | 91.7% | 100.0% | 59.4% | 65.1% | 0.0% | 100.0% | 99.1% | 70.5% | 67.3% | 0.0% |
| $d = 36$ features | ↳ % At Least 1 Feature Responsive | 8.3% | 0.0% | 40.6% | 34.9% | 100.0% | 0.0% | 0.9% | 29.5% | 32.7% | 100.0% |
| $d_A = 9$ mutable | ↳ % All Features Responsive | 0.0% | 0.0% | 0.0% | 0.0% | **100.0%** | 0.0% | 0.0% | 0.0% | 0.0% | **100.0%** |
| Dua & Graff [15] | ↳ # Features Highlighted | 4.0 | 4.0 | 4.0 | 4.0 | 1.8 | 4.0 | 4.0 | 4.0 | 4.0 | 1.8 |
| givemecredit | % Presented with Explanations | 100.0% | 100.0% | 100.0% | 100.0% | 72.4% | 100.0% | 100.0% | 100.0% | 100.0% | 76.0% |
| $n = 120,268$ | ↳ % All Features Unresponsive | 65.5% | 46.8% | 33.1% | 56.0% | 0.0% | 41.8% | 43.3% | 31.6% | 30.6% | 0.0% |
| $d = 23$ features | ↳ % At Least 1 Feature Responsive | 34.5% | 53.2% | 44.0% | 66.9% | 100.0% | 58.2% | 56.7% | 68.4% | 69.4% | 100.0% |
| $d_A = 13$ mutable | ↳ % All Features Responsive | 0.0% | 0.0% | 0.0% | 22.8% | **100.0%** | 0.0% | 0.0% | 4.2% | 13.2% | **100.0%** |
| Kaggle [32] | ↳ # Features Highlighted | 4.0 | 4.0 | 4.0 | 4.0 | 2.4 | 4.0 | 4.0 | 4.0 | 4.0 | 2.6 |

**Table 4:** Responsiveness of feature-highlighting explanations for LR and XGB models for all methods and datasets. We defer results for RF to Appendix C.1 for clarity. For each model, we generate explanations that highlight up to 4 top-scoring features from a given method. We report the proportion of individuals receiving an explanation (*% Presented with Explanations*) and the mean number of features in each explanation (*# Features Highlighted*). We also show the proportion of instances where all features are unresponsive (*% All Features Unresponsive*) highlighting positive values; at least one feature is responsive (*% At Least 1 Feature Responsive*), or all features are responsive (*% All Features Responsive*) highlighting the **best value**.

on the heloc dataset, we find that 92.7% and 77.3% of explanations from LIME and SHAP fail to include a single responsive feature. This stems from two issues:

- *Low Scores for Responsive Features*: Under the LR model on the heloc dataset, for example, 41.0% of denied individuals can achieve recourse by altering a single feature. However, LIME and SHAP do not assign higher scores to these features (see plots in Appendix C.2).

- *Fixed Predictions*: Under the LR model on the heloc dataset, 22.2% of denied individuals are assigned a fixed prediction. These are instances where providing any explanation can be harmful. However, LIME, SHAP and their variants still highlight important features. For example, a SHAP explanation for a fixed prediction includes AvgYearsInFile and NetFractionRevolvingBurden, which gives the impression that the individual could change them to attain recourse.

Provided that feature attribution methods like LIME and SHAP can output unresponsive features by overlooking actionability constraints, we study the potential to improve responsiveness using their *action-aware* variants SHAP-AW and LIME-AW. We construct explanations using only actionable features, following a common belief that we can account for actionability by post-processing [e.g., 43]. Our results in Table 4 show that action-aware variants can highlight more responsive features. For the LR model in heloc, 29.7% of SHAP-AW explanations contained at least one responsive feature (c.f., 22.7% for SHAP), meaning that consumers are more likely to be informed of a feature that can lead to recourse. One shortcoming of this approach is that we are required to filter features based on their actionability at a global level (e.g., whether individuals can intervene on this feature in principle). This is the most conservative approach in accounting for actionability, which does not require additional assumptions about constraints at an individual-level.

**On Explanations with Responsiveness Scores** Our results in Table 4 show how our approach can be used to comply with regulatory requirements and address the two limitations that affect feature attribution methods. When we construct feature-highlighting explanations using responsiveness scores, we present individuals with explanations that only contain responsive features, achieving 100% on the *% All Features Responsive* metric across datasets and models. In contrast, only 0.2% of SHAP-AW explanations of the LR model in heloc were fully responsive. For the remaining 99.8%, we are guaranteed that the explanation contains at least one unresponsive feature, potentially leading individuals to intervene on them and still fail to attain recourse.

Furthermore, explanations based on responsiveness scores contain the *most* responsive features that can be changed independently to obtain recourse. In effect, we only provide explanations to

individuals who can achieve recourse with a single-feature action. This may result in explanations that highlight fewer features on average – for example, individuals receiving explanations from the LR model on `heloc` receive 2.3 out of 4 reasons on average. This behavior can mitigate harm as we avoid presenting explanations to individuals with fixed predictions or those who require joint actions to change their outcomes.

## 5  DISCUSSION

Our results in Section 4 show that feature attribution methods fail to highlight responsive features. In principle, explanations should highlight features like `income`, that are responsive, that will lead to a desired prediction when altered past (or below) a certain threshold, and that most people know they should increase (or decrease). In practice, this may not be the case; features may have to be changed in ways that are not monotonic (e.g., increase `savings` by at least $1,000 but not more than $2,500) or not intuitive (i.e., decrease `income` to be approved for a loan). In what follows, we evaluate how often explanations highlight features that meet these additional conditions. Our goal is to show how such analyses can support the kind of information we must include in an explanation.

**Setup**   We work with a version of the `givemecredit` dataset where we keep continuous features as-is rather than binarize them (Appendix D). We use the same setup as in Section 4 to train an XGB model and build feature-highlighting explanations using SHAP and LIME. We evaluate the responsiveness of features using reachable sets built using Algorithm 1: for each individual with an adverse prediction under the XGB model ($n = 23, 459$), we constructed a sampled reachable set for each actionable feature. We chose the sample size $N = 500$ to ensure that the $100(1 - \alpha)\%$ confidence interval in Appendix A.2 had an upper bound $\leq 0.01$ when $\hat{\mu}_j(\boldsymbol{x}) = 0$ with $\alpha = 0.01$.

**On the Pitfalls of Missing Information**   Our results reveal how individuals who are shown a list of responsive features can fail to attain a desired prediction due to missing information.

- *Missing Information on Degree of Change*: Explanations can often highlight features without including information on how much to change them. As an example, 32.3% of SHAP explanations highlight `CreditUtilization`, the proportion of available credit in use. In this case, the feature is responsive in 10.7% explanations. However, 9% of individuals would only be approved if they change this feature by specific amounts. As an example, we can point to an individual with `CreditUtilization` $= 0.99$ who would only be approved by reducing this value to $x_j \in (0, 0.504) \cup (0.651, 0.677)$ – i.e., they would be approved if they were to reduce to 0.66. but not to 0.55. This is an instance where responsiveness isn't *monotonic* – `CreditUtilization` is responsive in disjoint intervals.

- *Missing Information on Direction of Change*: Explanations can also highlight features without stating whether they must be increased or decreased. This can backfire when features have to be changed in counterintuitive ways. As an example, we consider `MonthlyIncome`, which is highlighted in 48.1% of SHAP explanations. In this case, the feature is responsive in 20.6% of them. However, 10.7% of cases would only lead to approval if individuals were to decrease `MonthlyIncome`. For example, we had an applicant with `MonthlyIncome` $= 4100$, that could get approved for a loan but only if they were to *decrease* it by 581 or more.

We show how these pitfalls affect explanations from SHAP and LIME in Table 5. Our results show that less than 7% of explanations highlight at least one feature that is responsive, monotonic and intuitive. However, 0% of people are given feature-highlighting explanations where all features meet these conditions. Hence, at least one feature in each explanation violate these conditions. These results broadly highlight that there is no regime where an individual can reliably achieve recourse using a standard feature-highlighting explanation, and underscore the need to include additional information when explanations are meant to support recourse.

**Discussion**   One of the consequences of building explanations using methods like SHAP and LIME is that we cannot reliably tell when these conditions are met. We could impose some of these conditions by enforcing conditions on how we train the model – i.e., we could ensure monotonicity by using a linear classifier like LR rather than XGB. Alternatively, we could use custom responsiveness scores to highlight features that meet all these conditions (i.e., is responsive, monotonic, and intuitive). RESP

column in Table 5 show how effective custom responsiveness scores can be as a standalone solution. It can highlight instances where there are no features that meet the these conditions (i.e., the "0" bar) – at which point we could highlight features that achieve weaker forms of responsiveness alongside additional information to help consumers. For example, if we find features that are responsive but non-monotonic, then we may add additional information on the thresholds. This would provide an alternative approach to ensure these conditions in a way that would not interfere with model development.

## 6 CONCLUDING REMARKS

Explanations are essential for consumer safeguards in domains like lending and hiring, providing critical information that enables consumers to exercise their rights [16]. We demonstrate how explanations can backfire by highlighting unresponsive features and by explaining fixed predictions. Our work addresses these issues by constructing explanations based on feature *responsiveness* – the probability that a feature can be changed to attain a desired prediction. Our model-agnostic framework can readily be used in place of methods like SHAP and LIME, which are currently being used to comply with regulations. In doing so, we can strengthen consumer protection by highlighting responsive features and flagging instances where is no recourse.

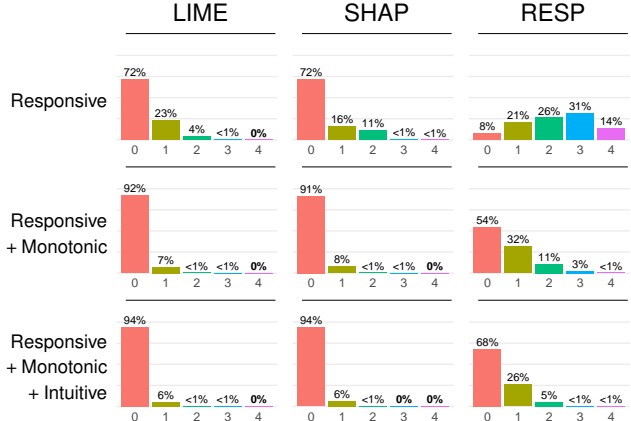

**Table 5:** Distribution of the number of features in feature-highlighting explanations from LIME, SHAP and RESP for XGB model on `givemecredit` that are: responsive (can change independently to obtain recourse), monotonic (consistently responsive past a threshold) and intuitive (responsive in a direction that aligns with common expectations). We quantify to what extent explanations highlight features that can reliably provide recourse without additional information. We evaluate these characteristics using the feature's reachable set $\hat{R}_j(\boldsymbol{x})$ (see Appendix D).

Our results also underscore key lessons for how to design, enforce, and comply with explainability mandates. Specifically, we show that these mandates should establish clearer guidelines pertaining to the content of explanations and how they are constructed so that explanations serve their outlined objectives – i.e., require companies to use tailored solutions that support recourse. This is especially important as many of the other desiderata for explanations have standalone solutions – e.g., anti-discrimination via auditing [49, 6] and less discriminatory models [29, 63].

**Limitations**   The main limitations of our work stem from assumptions about actionability. In this work, we have used a conservative set of assumptions that reflect indisputable constraints. In effect, responsiveness scores under these constraints can flag individuals with fixed predictions. However, we may not guarantee recourse as individuals may face additional constraints that we are not aware of. In practice, we can mitigate these issues by highlighting additional features, reporting features that exceed a minimal level of responsiveness, or eliciting constraints from decision subjects [see e.g., 18, 12, 37]. We have also restricted our attention to settings where interventions lead to deterministic causal effects on downstream features. In principle, our machinery can handle settings where interventions induce probabilistic effects on downstream features [35, 14, 59]: given an individual probabilistic graphical model, we can compute a responsiveness score reflecting the expected recourse rate. Nevertheless, these causal assumptions are difficult to validate at an individual level and thus require an approach to measure responsiveness in a way that is robust to misspecification.

**Broader Applications**   Our framework for measuring feature responsiveness extends to various contexts beyond consumer finance. More broadly, responsiveness scores can evaluate model behavior in relation to user inputs or interactions. For instance, in content moderation, these scores can assess a model's robustness against strategic manipulations. In criminal justice applications, they can determine whether a model responds inappropriately to changes in protected attributes (e.g., sex) or appropriately to relevant factors (e.g., type of criminal offense).

ACKNOWLEDGEMENTS

This work is supported by the National Science Foundation under award IIS-2313105, and the NIH Bridge2AI Center Grant U54HG012510.

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

# Supplementary Materials

## A   SUPPLEMENTARY MATERIAL FOR SECTION 3

Define a partition $\{\pi_1, \pi_2, \ldots, \pi_k\}$ of $[d]$ such that given two parts $\pi_m, \pi_n$, there are no joint constraints between all pairs $(p, q) \in \pi_m \times \pi_n$ of features. Another way to think about feature partitions would be as connected components in a graph, where features are nodes and edges represent joint constraints (i.e., $\exists$ edge $(p, q) \iff$ there are joint actionability constraints between $p$ and $q$).

Let $\pi'$ be a part such that $j \in \pi'$.

### A.1   IMPLEMENTATION DETAILS FOR REACHABLE SET ENUMERATION

**Description of** Find1DAction **Routine**   The Find1DAction routine in Algorithm 2 enumerates a set of possible actions given an intervention on a feature by repeatedly solving a mixed-integer program. At each iteration, the routine takes as input:

- $\boldsymbol{x} \in \mathcal{X}$, a point
- $\boldsymbol{v} \in V_j(\boldsymbol{x})$, a valid intervention on feature $j$
- $A_j(\boldsymbol{x})$, the action set for feature $j$ (which is specified by separable and joint actionability constraints on $j$)
- $\mathcal{A}_j^{\text{opt}}$, a set of $[L]$ actions enumerated over previous iterations.

Given these inputs, the procedure solves the following mixed-integer program in Eq. (3) to return the nearest single-feature actions from the set $\boldsymbol{a} \in A_j(\boldsymbol{x}) \setminus \mathcal{A}_j^{\text{opt}}$ when they exist, or to confirm that there are no more actions to enumerate.

$$\min_{\boldsymbol{a}} \quad \sum_{k \in \pi'} a_k^+ + a_k^-$$

$$
\begin{aligned}
\text{s.t.} \quad & a_j = v_j & & \textit{intervene with } \boldsymbol{v} & \text{(3a)} \\
& a_k = a_k^+ - a_k^- & k \in \pi' & \textit{reconstruction of } a_k & \text{(3b)} \\
& a_k^+ \geq a_k & k \in \pi' & \textit{positive component of } a_k & \text{(3c)} \\
& a_k^- \geq -a_k & k \in \pi' & \textit{negative component of } a_k & \text{(3d)} \\
& a_k^+ \leq |\sup V_k(\boldsymbol{x})|\sigma_k & k \in \pi' & a_k^+ > 0 \implies \sigma_k = 1 & \text{(3e)} \\
& a_k^- \leq |\inf V_k(\boldsymbol{x})|(1 - \sigma_k) & k \in \pi' & a_k^- > 0 \implies \sigma_k = 0 & \text{(3f)} \\
& a_k = a_{k,l} + \delta_{k,l}^+ - \delta_{k,l}^- & k \in \pi', \boldsymbol{a}_l \in \mathcal{A}_j^{\text{opt}} & \textit{maintain distance from prior actions} & \text{(3g)} \\
& \varepsilon_{\min} \leq \sum_{k \in \pi'}(\delta_{k,l}^+ + \delta_{k,l}^-) & \boldsymbol{a}_l \in \mathcal{A}_j^{\text{opt}} & \textit{any solution is } \varepsilon_{min} \textit{ away from } \boldsymbol{a}_l & \text{(3h)} \\
& \delta_{k,l}^+ \leq M_{k,l}^+ u_{k,l} & k \in \pi', l \in [L] & \delta_{k,l}^+ > 0 \implies u_{k,l} = 1 & \text{(3i)} \\
& \delta_{k,l}^- \leq M_{k,l}^-(1 - u_{k,l}) & k \in \pi', l \in [L] & \delta_{k,l}^- > 0 \implies u_{k,l} = 0 & \text{(3j)} \\
& \boldsymbol{a} \in A_j(\boldsymbol{x}) & & \textit{joint actionability constraints on } j & \text{(3k)} \\
& a_k^+, a_k^- \in \mathbb{R}_+ & k \in \pi' & \textit{absolute value of } a_k & \text{(3l)} \\
& \delta_{k,l}^+, \delta_{k,l}^- \in \mathbb{R}_+ & k \in \pi', l \in [L] & \textit{signed distances from } a_{k,l} & \text{(3m)} \\
& u_{k,l} \in \{0, 1\} & k \in \pi', l \in [L] & \textit{sign indicator of } \delta_{k,l} & \text{(3n)} \\
& \sigma_k \in \{0, 1\} & k \in \pi' & \textit{sign indicator of } a_k & \text{(3o)}
\end{aligned}
$$

As mentioned in Section 3.1, Find1DAction solves the following optimization problem:

$$\textsf{Find1DAction}(\boldsymbol{x}, \boldsymbol{v}, A_j) := \operatorname*{argmin}_{\boldsymbol{a} \in A_j(\boldsymbol{x})} \|\boldsymbol{a}\|_1 \text{ s.t. } a_j = v_j$$

Since $\|\boldsymbol{a}\|_1 = \sum_{k \in [d]} |a_k|$, we model the absolute value using positive and negative components $a_k^+, a_k^-$ (constraints (3b), (3l)). In similar fashion, we decompose $\delta_{k,l}$, which is the distance from existing solutions $\boldsymbol{a}_l \in \mathcal{A}_j^{\text{opt}}$ into $\delta_{k,l}^+, \delta_{k,l}^-$ (constraints (3g), (3m)). $\sigma_k$ and $u_{k,l} \in \{0, 1\}$ are indicator

variables for the sign of $a_k$ and $\delta_{k,l}$, where 1 indicates its positive and 0 otherwise (constraint (3n), (3o)).

The first constraint, (3a) enforces that we intervene on feature $j$ with $v$. The remaining constraints describe three key requirements for $a$:

1. Sufficient distance from prior solutions (constraint (3h))

2. Adherence to separable actionability constraints (constraint (3e), (3f), (3i), (3j))

3. Adherence to joint actionability constraints (constraint (3k))

Constraint (3h) ensures that given $\varepsilon_{\min} > 0$, $\|a - a_l\|_1 \geq \varepsilon_{\min} \ \forall \, a_l \in \mathcal{A}_j^{\text{opt}}$. We set $\varepsilon_{\min} = 0.5$ for our experiments with discrete datasets.

Constraints (3e), (3f) ensure that $a_k$ is feasible under separable constraints on $k \in \pi'$ and that only one of $a_k^+$ or $a_k^-$ is strictly positive. Similarly, constraints (3i), (3j) ensure that the distances between $a$ and each $a_l$ are within some bound. We achieve this by setting "Big-M" parameters $M_{k,l}^+, M_{k,l}^-$, which represent the upper bound for $\delta_{k,l}^+$ and $\delta_{k,l}^-$. For each feature $k \in \pi'$, we let

$$M_{k,l}^+ := |\sup V_k(x) - a_{k,l}|, \ M_{k,l}^- := |\inf V_k(x) - a_{k,l}|$$

Along with the indicator variable $u_{k,l}$, $M_{k,l}^+, M_{k,l}^-$ ensure that only one of $\delta_{k,l}^+$ or $\delta_{k,l}^-$ is strictly positive and is feasible under separable actionability constraints.

Constraint (3k) ensures that $a$ also adheres to joint actionability constraints. These constraints will exist if and only if $|\pi'| > 1$. See [38] for examples of how we can explicitly encode joint actionability constraints into Eq. (3).

This formulation is an adaptation of the MIP presented in [38]. Our formulation above differs in that we solve for actions with respect to a fixed intervention $v$.

## A.2 Implementation Details for Reachable Set Sampling

Sampling a reachable set requires additional considerations.

**Choosing a Sample Size** The sample size $N$ controls the precision of the estimated responsiveness score $\hat{\mu}_j(x)$. We formalize precision using confidence intervals by treating $\hat{\mu}_j(x)$ as a binomial distribution parameter:

**Remark 2.** *Given a point $x \in \mathcal{X}$, let $\hat{R}_j(x)$ denote a sample of $N$ points drawn uniformly from the reachable set $R_j(x)$. Given any model $f : \mathcal{X} \to \mathcal{Y}$, we can estimate the responsiveness score for feature $j$ as $\hat{\mu}_j(x) := \frac{1}{N} \sum_{x' \in \hat{R}_j(x)} \mathbb{1}[f(x') = \hat{y}^{target}]$. Given a significance level $\alpha \in (0,1)$, we have that:*

$$\Pr(\mu_j(x) \in [\tilde{\mu}_j(x) - \mathcal{E}, \tilde{\mu}_j(x) + \mathcal{E}]) \geq 1 - \alpha$$

*Here: $\mathcal{E} := \kappa \sqrt{\frac{1}{N+\kappa^2} \tilde{\mu}_j(x)(1 - \tilde{\mu}_j(x))}$ and $\tilde{\mu}_j(x) := \frac{1}{N+\kappa^2}\left(S + \frac{\kappa^2}{2}\right)$ is a corrected estimator to improve coverage when $\mu_j(x) \in \{0,1\}$ [10], $S := |\{x' \in \hat{R}_j(x) \mid f(x') = \hat{y}^{target}\}|$ is the subset of responsive points, and $\kappa := \Phi^{-1}(1 - \frac{\alpha}{2})$ is a constant based on the Normal CDF $\Phi(\cdot)$.*

The Agresti–Coull interval above is an approximate confidence interval for a binomial proportion [3], offering an improvement over the standard normal approximation known as the Wald Interval. It is particularly effective for small proportion values, providing more reliable coverage – the probability that the interval contains the true parameter value [10].

**Fact 3.** *For a fixed $\alpha$, $N$, $\mathcal{E}$ is maximized when $S = \frac{N}{2}$ and attains its minimum at $S = 0$ and $S = N$.*

*Proof.* Let $z = \frac{S + \frac{\kappa^2}{2}}{N + \kappa^2}$. Then, we have:

$$\mathcal{E} = \kappa \sqrt{\frac{z(1-z)}{(N + \kappa^2)}}$$

Since $N$ and $\alpha$ are fixed ($\kappa$ is defined by $\alpha$), $\mathcal{E}$ is a function of $z$ of the form $h(z) = c\sqrt{z(1-z)}$ where $c \in \mathbb{R}^+$.

Take the derivative with respect to $w$:

$$h'(z) = c \cdot \frac{1}{2}, [z(1-z)]^{-\frac{1}{2}}(1-2z)$$
$$= \frac{c(1-2z)}{2\sqrt{z(1-z)}}.$$

See that $h'(z) = 0$ when $z = \frac{1}{2}$

Take the second derivative to check that it is concave.

$$h''(z) = \frac{c}{2}\left[-2[z(1-z)]^{-\frac{1}{2}} - \frac{1}{2}(1-2z)^2[z(1-z)]^{-\frac{3}{2}}\right]$$
$$= -c\frac{z(1-z) + \frac{1}{4}(1-2z)^2}{[z(1-z)]^{\frac{3}{2}}}$$
$$= -\frac{c}{4}[z(1-z)]^{-\frac{3}{2}}$$

Since $z > 0$ and $c > 0$, $h''(z) < 0$, hence it is concave and attains its maximum when $z = \frac{1}{2} \iff S = \frac{N}{2}$.

We see that $h'(z) > 0$ where $z < \frac{1}{2}$, meaning it is increasing for $z \in (0, \frac{1}{2}]$. Hence the local minimum is achieved at the smallest possible $z$ – when $S = 0$.

Similarly, for $z \in [\frac{1}{2}, 1)$, $h'(z) < 0$ and the local minimum is achieved at the largest possible $z$ – when $S = N$.

Note that the value of $h$ (or $\mathcal{E}$) are the same at those two points.

$\square$

Using the fact above, we list two ways of setting $N$ (given $\alpha$):

1. Control the precision when $S = 0$ (i.e., no points in $\hat{R}_j(\boldsymbol{x})$ are responsive) $\iff$ control the width of the shortest interval

2. Control the precision when $S = \frac{N}{2}$ (i.e., half of the points in $\hat{R}_j(\boldsymbol{x})$ are responsive) $\iff$ control the width of the widest interval

Either way, we fix $\alpha$ and solve for $N$ given the width of the interval $\mathcal{E}$ at a specified $S$. Below we provide a table of the smallest $N$ required for different $\mathcal{E}$ – interval widths – at common values of $\alpha$ for the two methods:

| $\alpha$ | $\mathcal{E}$ | | | |
|---|---|---|---|---|
| | 0.01 | 0.02 | 0.05 | 0.10 |
| 0.01 | 461 | 227 | 86 | 39 |
| 0.05 | 267 | 132 | 50 | 23 |
| 0.10 | 188 | 93 | 35 | 16 |

| $\alpha$ | $\mathcal{E}$ | | | |
|---|---|---|---|---|
| | 0.01 | 0.02 | 0.05 | 0.10 |
| 0.01 | 16581 | 4141 | 657 | 160 |
| 0.05 | 9600 | 2398 | 381 | 93 |
| 0.10 | 6762 | 1689 | 268 | 65 |

**Table 6:** Minimum $N$ required to ensure the shortest confidence interval is less than $2\mathcal{E}$ (Method 1)

**Table 7:** Minimum $N$ required to ensure the widest confidence interval is less than $2\mathcal{E}$ (Method 2)

**Description of the** Sample1DAction **Routine** Let $j$ be the feature that we are intervening on.

**Case 1:** $|\pi'| = 1$ (i.e., $\pi' = \{j\}$, $j$ is not jointly constrained with other features).

Here, there are no downstream effects from intervening on $j$. We take a uniformly random intervention from $V_j(\boldsymbol{x})$:

$$\boldsymbol{a}^* \sim V_j(\boldsymbol{x})$$

which abides by $j$'s separable actionability constraints like feature bounds and monotonicity.

**Case 2:** $|\pi'| > 1$ (i.e., $j$ is jointly constrained with other features)

Similarly to **Case 1**, we first proceed by taking a uniformly random intervention on $j$:

$$\boldsymbol{v} \sim V_j(\boldsymbol{x})$$

However, there are downstream effects on the intervention. We breakdown $\pi'$ into three disjoint subsets:

$$\pi' = \{j\} \cup \pi'_{\text{disc}} \cup \pi'_{\text{cts}}$$

where $\pi'_{\text{disc}}$ and $\pi'_{\text{cts}}$ are the sets of discrete and and continuous features in $\pi'$ respectively.

We consider the following three sub-cases:

*Case 2a:* $|\pi'_{\text{cts}}| = 0$ – all other features in $\pi'$ are discrete.

We repeatedly solve the MIP in Find1DAction to build $S(\boldsymbol{v}) = \{\boldsymbol{a} \in A_j(\boldsymbol{x}) \mid a_j = v_j\}$, the set of all possible actions resulting from intervention $\boldsymbol{v}$. The resulting action is a uniformly random sample from $S(\boldsymbol{v})$: $\boldsymbol{a}^* \sim S(\boldsymbol{v})$

*Case 2b:* $|\pi'_{\text{disc}}| = 0$ – all other features in $\pi'$ are continuous.

We sample interventions for each feature in $\pi'_{\text{disc}}$:

$$\boldsymbol{z}(\boldsymbol{v}) = \sum_{k \in \pi'_{\text{cts}}} \boldsymbol{z}_k$$

where $\boldsymbol{z}_k \sim V_k(\boldsymbol{x})$, a sample from the intervention set for feature $k$.

Then we check feasibility for $\boldsymbol{a}^* = \boldsymbol{v} + \boldsymbol{z}(\boldsymbol{v})$ by running CheckFeasibility($\boldsymbol{a}^*, A_j(\boldsymbol{x})$).

We repeat until we find a feasible $\boldsymbol{a}^*$.

*Case 2c:* $|\pi'_{\text{cts}}|, |\pi'_{\text{disc}}| > 0$ – part contains discrete and continuous features.

We run the sampling steps in *Case 2a, 2b* for $\pi'_{\text{disc}}$ and $\pi'_{\text{cts}}$ to get $\boldsymbol{a}_{\text{disc}}$ and $\boldsymbol{a}_{\text{cts}}$.

We then check feasibility on $\boldsymbol{a}^* = \boldsymbol{a}_{\text{disc}} + \boldsymbol{a}_{\text{cts}} - \boldsymbol{v}$ (we subtract $\boldsymbol{v}$ since both $\boldsymbol{a}_{\text{disc}}, \boldsymbol{a}_{\text{cts}}$ contain $\boldsymbol{v}$) by running CheckFeasibility($\boldsymbol{a}^*, A_j(\boldsymbol{x})$).

We repeat until we find a feasible $\boldsymbol{a}^*$.

**Description of** CheckFeasibility **Routine** We describe the implementation for the CheckFeasibility($\boldsymbol{x}, \boldsymbol{a}^*, A_j$) in Algorithm 1. Contrary to the MIP formulation in Appendix A.1, given the original point $\boldsymbol{x} \in \mathcal{X}$ and the sampled action $\boldsymbol{a}^*$, we solve the MIP once.

$$
\begin{aligned}
\min_{\boldsymbol{a}} \quad & 1 \\
\text{s.t.} \quad & a_j = v_j && \textit{intervene with } \boldsymbol{v} && \text{(4a)} \\
& a_k = a_k^+ - a_k^- && k \in \pi' && \textit{reconstruction of } a_k && \text{(4b)} \\
& a_k^+ \geq a_k && k \in \pi' && \textit{positive component of } a_k && \text{(4c)} \\
& a_k^- \geq -a_k && k \in \pi' && \textit{negative component of } a_k && \text{(4d)} \\
& a_k^+ \leq |\sup V_k(\boldsymbol{x})|\sigma_k && k \in \pi' && a_k^+ > 0 \implies \sigma_k = 1 && \text{(4e)} \\
& a_k^- \leq |\inf V_k(\boldsymbol{x})|(1 - \sigma_k) && k \in \pi' && a_k^- > 0 \implies \sigma_k = 0 && \text{(4f)} \\
& \boldsymbol{a} \in A_j(\boldsymbol{x}) && && \textit{joint actionability constraints on } j && \text{(4g)} \\
& a_k^+, a_k^- \in \mathbb{R}_+ && k \in \pi' && \textit{absolute value of } a_k && \text{(4h)}
\end{aligned}
$$

$$\sigma_k \in \{0, 1\} \qquad k \in \pi' \quad \textit{sign indicator of } a_k \tag{4i}$$

The formulation is a variant of the problem in Appendix A.1, where:

- $\boldsymbol{a} = \boldsymbol{a}^*$,
- $\mathcal{A}_j^{\text{opt}} = \varnothing$,
- and set the objective to $\min_{\boldsymbol{a}} \ 1$

Hence $\mathsf{CheckFeasibility}(\boldsymbol{x}, \boldsymbol{a}^*, A_j) = 1$ if $\boldsymbol{a}^*$ is feasible under actionability constraints and 0 otherwise.

# B SUPPLEMENTARY EXPERIMENT DETAILS

## B.1 DETAILS FOR THE heloc DATASET

The heloc dataset was created to predict repayment on Home Equity Line of Credit HELOC applications; these are loans that use people's homes as collateral. The anonymized version of the dataset was developed by FICO for use in an Explainable Machine Learning Challenge in 2018 [23]. Each instance in the dataset is an application for a home equity loan from a single applicant. There are $n = 10,459$ instances and $d = 23$ features. Here, the label $y_i = 0$ if an applicant has been more than 90 days overdue on their payments in the last 2 years and $y_i = 1$ otherwise. We thermometer encode continuous or integer features after dropping rows and features with missing data (see Table 8). See GitHub for dataset processing code.

| Name | Type | LB | UB | Actionability | Sign | Joint Constraints | Partition ID |
|---|---|---|---|---|---|---|---|
| NumInstallTrades≥2 | {0,1} | 0 | 1 | Yes | + | 20, 21, 24, 25, 28, 29, 32, 33 | 14 |
| NumInstallTradesWBalance≥2 | {0,1} | 0 | 1 | Yes | + | 20, 21, 24, 25, 28, 29, 32, 33 | 14 |
| NumInstallTrades≥3 | {0,1} | 0 | 1 | Yes | + | 20, 21, 24, 25, 28, 29, 32, 33 | 14 |
| NumInstallTradesWBalance≥3 | {0,1} | 0 | 1 | Yes | + | 20, 21, 24, 25, 28, 29, 32, 33 | 14 |
| NumInstallTrades≥5 | {0,1} | 0 | 1 | Yes | + | 20, 21, 24, 25, 28, 29, 32, 33 | 14 |
| NumInstallTradesWBalance≥5 | {0,1} | 0 | 1 | Yes | + | 20, 21, 24, 25, 28, 29, 32, 33 | 14 |
| NumInstallTrades≥7 | {0,1} | 0 | 1 | Yes | + | 20, 21, 24, 25, 28, 29, 32, 33 | 14 |
| NumInstallTradesWBalance≥7 | {0,1} | 0 | 1 | Yes | + | 20, 21, 24, 25, 28, 29, 32, 33 | 14 |
| NumRevolvingTrades≥2 | {0,1} | 0 | 1 | Yes | + | 22, 23, 26, 27, 30, 31, 34, 35 | 15 |
| NumRevolvingTradesWBalance≥2 | {0,1} | 0 | 1 | Yes | + | 22, 23, 26, 27, 30, 31, 34, 35 | 15 |
| NumRevolvingTrades≥3 | {0,1} | 0 | 1 | Yes | + | 22, 23, 26, 27, 30, 31, 34, 35 | 15 |
| NumRevolvingTradesWBalance≥3 | {0,1} | 0 | 1 | Yes | + | 22, 23, 26, 27, 30, 31, 34, 35 | 15 |
| NumRevolvingTrades≥5 | {0,1} | 0 | 1 | Yes | + | 22, 23, 26, 27, 30, 31, 34, 35 | 15 |
| NumRevolvingTradesWBalance≥5 | {0,1} | 0 | 1 | Yes | + | 22, 23, 26, 27, 30, 31, 34, 35 | 15 |
| NumRevolvingTrades≥7 | {0,1} | 0 | 1 | Yes | + | 22, 23, 26, 27, 30, 31, 34, 35 | 15 |
| NumRevolvingTradesWBalance≥7 | {0,1} | 0 | 1 | Yes | + | 22, 23, 26, 27, 30, 31, 34, 35 | 15 |
| YearsOfAccountHistory | ℤ | 0 | 50 | No | | 5, 17, 18, 19 | 5 |
| YearsSinceLastDelqTrade≤1 | {0,1} | 0 | 1 | Yes | + | 5, 17, 18, 19 | 5 |
| YearsSinceLastDelqTrade≤3 | {0,1} | 0 | 1 | Yes | + | 5, 17, 18, 19 | 5 |
| YearsSinceLastDelqTrade≤5 | {0,1} | 0 | 1 | Yes | + | 5, 17, 18, 19 | 5 |
| NetFractionInstallBurden≥10 | {0,1} | 0 | 1 | Yes | + | 36, 37, 38 | 16 |
| NetFractionInstallBurden≥20 | {0,1} | 0 | 1 | Yes | + | 36, 37, 38 | 16 |
| NetFractionInstallBurden≥50 | {0,1} | 0 | 1 | Yes | + | 36, 37, 38 | 16 |
| NetFractionRevolvingBurden≥10 | {0,1} | 0 | 1 | Yes | + | 39, 40, 41 | 17 |
| NetFractionRevolvingBurden≥20 | {0,1} | 0 | 1 | Yes | + | 39, 40, 41 | 17 |
| NetFractionRevolvingBurden≥50 | {0,1} | 0 | 1 | Yes | + | 39, 40, 41 | 17 |
| AvgYearsInFile≥3 | {0,1} | 0 | 1 | Yes | + | 6, 7, 8 | 6 |
| AvgYearsInFile≥5 | {0,1} | 0 | 1 | Yes | + | 6, 7, 8 | 6 |
| AvgYearsInFile≥7 | {0,1} | 0 | 1 | Yes | + | 6, 7, 8 | 6 |
| MostRecentTradeWithinLastYear | {0,1} | 0 | 1 | Yes | | 9, 10 | 7 |
| MostRecentTradeWithinLast2Years | {0,1} | 0 | 1 | Yes | | 9, 10 | 7 |
| ExternalRiskEstimate≥40 | {0,1} | 0 | 1 | No | | – | 0 |
| ExternalRiskEstimate≥50 | {0,1} | 0 | 1 | No | | – | 1 |
| ExternalRiskEstimate≥60 | {0,1} | 0 | 1 | No | | – | 2 |
| ExternalRiskEstimate≥70 | {0,1} | 0 | 1 | No | | – | 3 |
| ExternalRiskEstimate≥80 | {0,1} | 0 | 1 | No | | – | 4 |
| AnyDerogatoryComment | {0,1} | 0 | 1 | No | | – | 8 |
| AnyTrade120DaysDelq | {0,1} | 0 | 1 | No | | – | 9 |
| AnyTrade90DaysDelq | {0,1} | 0 | 1 | No | | – | 10 |
| AnyTrade60DaysDelq | {0,1} | 0 | 1 | No | | – | 11 |
| AnyTrade30DaysDelq | {0,1} | 0 | 1 | No | | – | 12 |
| NoDelqEver | {0,1} | 0 | 1 | No | | – | 13 |
| NumBank2Natl TradesWHighUtilizationGeq2 | {0,1} | 0 | 1 | Yes | + | – | 18 |

**Table 8:** Separable Actionability Constraints for the processed heloc dataset. **Type** indicates the feature type (ℤ for integer, {0,1} for binary). **LB**, **UB** are the lower and upper bounds for the feature. **Actionability** indicates whether the feature is globally actionable. **Sign** indicates if the feature can only increase (+) or decrease (-). **Joint Constraints** are a list non-separable constraint indices it is tied to (if any). **Partition ID** indicates which partition the feature belongs to.

**Actionability Constraints**  The joint actionability constraints listed in Table 8 include:

1. DirectionalLinkage: Actions on NumRevolvingTradesWBalance≥2 will induce to actions on ['NumRevolvingTrades≥2'].Each unit change in NumRevolvingTradesWBalance≥2 leads to:1.00-unit change in NumRevolvingTrades≥2

2. DirectionalLinkage: Actions on NumInstallTradesWBalance≥2 will induce to actions on ['NumInstallTrades≥2'].Each unit change in NumInstallTradesWBalance≥2 leads to:1.00-unit change in NumInstallTrades≥2

3. DirectionalLinkage: Actions on NumRevolvingTradesWBalance≥3 will induce to actions on ['NumRevolvingTrades≥3'].Each unit change in NumRevolvingTradesWBalance≥3 leads to:1.00-unit change in NumRevolvingTrades≥3

4. DirectionalLinkage: Actions on `NumInstallTradesWBalance`$\geq$3 will induce to actions on ['`NumInstallTrades`$\geq$3'].Each unit change in `NumInstallTradesWBalance`$\geq$3 leads to:1.00-unit change in `NumInstallTrades`$\geq$3

5. DirectionalLinkage: Actions on `NumRevolvingTradesWBalance`$\geq$5 will induce to actions on ['`NumRevolvingTrades`$\geq$5'].Each unit change in `NumRevolvingTradesWBalance`$\geq$5 leads to:1.00-unit change in `NumRevolvingTrades`$\geq$5

6. DirectionalLinkage: Actions on `NumInstallTradesWBalance`$\geq$5 will induce to actions on ['`NumInstallTrades`$\geq$5'].Each unit change in `NumInstallTradesWBalance`$\geq$5 leads to:1.00-unit change in `NumInstallTrades`$\geq$5

7. DirectionalLinkage: Actions on `NumRevolvingTradesWBalance`$\geq$7 will induce to actions on ['`NumRevolvingTrades`$\geq$7'].Each unit change in `NumRevolvingTradesWBalance`$\geq$7 leads to:1.00-unit change in `NumRevolvingTrades`$\geq$7

8. DirectionalLinkage: Actions on `NumInstallTradesWBalance`$\geq$7 will induce to actions on ['`NumInstallTrades`$\geq$7'].Each unit change in `NumInstallTradesWBalance`$\geq$7 leads to:1.00-unit change in `NumInstallTrades`$\geq$7

9. DirectionalLinkage: Actions on `YearsSinceLastDelqTrade`$\leq$1 will induce to actions on ['`YearsOfAccountHistory`'].Each unit change in `YearsSinceLastDelqTrade`$\leq$1 leads to:-1.00-unit change in `YearsOfAccountHistory`

10. DirectionalLinkage: Actions on `YearsSinceLastDelqTrade`$\leq$3 will induce to actions on ['`YearsOfAccountHistory`'].Each unit change in `YearsSinceLastDelqTrade`$\leq$3 leads to:-3.00-unit change in `YearsOfAccountHistory`

11. DirectionalLinkage: Actions on `YearsSinceLastDelqTrade`$\leq$5 will induce to actions on ['`YearsOfAccountHistory`'].Each unit change in `YearsSinceLastDelqTrade`$\leq$5 leads to:-5.00-unit change in `YearsOfAccountHistory`

12. ReachabilityConstraint: The values of [`MostRecentTradeWithinLastYear`, `MostRecentTradeWithinLast2Years`] must belong to one of 4 values with custom reachability conditions.

13. ThermometerEncoding: Actions on [`YearsSinceLastDelqTrade`$\leq$1, `YearsSinceLastDelqTrade`$\leq$3, `YearsSinceLastDelqTrade`$\leq$5] must preserve thermometer encoding of YearsSinceLastDelqTradeleq., which can only decrease.Actions can only turn off higher-level dummies that are on, where `YearsSinceLastDelqTrade`$\leq$1 is the lowest-level dummy and `YearsSinceLastDelqTrade`$\leq$5 is the highest-level-dummy.

14. ThermometerEncoding: Actions on [`AvgYearsInFile`$\geq$3, `AvgYearsInFile`$\geq$5, `AvgYearsInFile`$\geq$7] must preserve thermometer encoding of AvgYearsInFilegeq., which can only increase.Actions can only turn on higher-level dummies that are off, where `AvgYearsInFile`$\geq$3 is the lowest-level dummy and `AvgYearsInFile`$\geq$7 is the highest-level-dummy.

15. ThermometerEncoding: Actions on [`NetFractionRevolvingBurden`$\geq$10, `NetFractionRevolvingBurden`$\geq$20, `NetFractionRevolvingBurden`$\geq$50] must preserve thermometer encoding of NetFractionRevolvingBurdengeq., which can only decrease.Actions can only turn off higher-level dummies that are on, where `NetFractionRevolvingBurden`$\geq$10 is the lowest-level dummy and `NetFractionRevolvingBurden`$\geq$50 is the highest-level-dummy.

16. ThermometerEncoding: Actions on [`NetFractionInstallBurden`$\geq$10, `NetFractionInstallBurden`$\geq$20, `NetFractionInstallBurden`$\geq$50] must preserve thermometer encoding of NetFractionInstallBurdengeq., which can only decrease.Actions can only turn off higher-level dummies that are on, where `NetFractionInstallBurden`$\geq$10 is the lowest-level dummy and `NetFractionInstallBurden`$\geq$50 is the highest-level-dummy.

17. ThermometerEncoding: Actions on [`NumRevolvingTradesWBalance`$\geq$2, `NumRevolvingTradesWBalance`$\geq$3, `NumRevolvingTradesWBalance`$\geq$5, `NumRevolvingTradesWBalance`$\geq$7] must preserve thermometer encoding of NumRevolvingTradesWBalancegeq., which can only decrease.Actions can only turn off higher-level dummies that are on, where `NumRevolvingTradesWBalance`$\geq$2 is the lowest-level dummy and `NumRevolvingTradesWBalance`$\geq$7 is the highest-level-dummy.

18. ThermometerEncoding: Actions on [`NumRevolvingTrades≥2`, `NumRevolvingTrades≥3`, `NumRevolvingTrades≥5`, `NumRevolvingTrades≥7`] must preserve thermometer encoding of NumRevolvingTradesgeq., which can only decrease.Actions can only turn off higher-level dummies that are on, where `NumRevolvingTrades≥2` is the lowest-level dummy and `NumRevolvingTrades≥7` is the highest-level-dummy.

19. ThermometerEncoding: Actions on [`NumInstallTradesWBalance≥2`, `NumInstallTradesWBalance≥3`, `NumInstallTradesWBalance≥5`, `NumInstallTradesWBalance≥7`] must preserve thermometer encoding of NumInstall-TradesWBalancegeq., which can only decrease.Actions can only turn off higher-level dummies that are on, where `NumInstallTradesWBalance≥2` is the lowest-level dummy and `NumInstallTradesWBalance≥7` is the highest-level-dummy.

20. ThermometerEncoding: Actions on [`NumInstallTrades≥2`, `NumInstallTrades≥3`, `NumInstallTrades≥5`, `NumInstallTrades≥7`] must preserve thermometer encoding of NumInstallTradesgeq., which can only decrease.Actions can only turn off higher-level dummies that are on, where `NumInstallTrades≥2` is the lowest-level dummy and `NumInstallTrades≥7` is the highest-level-dummy.

## B.2 DETAILS FOR THE german DATASET

The german dataset was created in 1994 and contains information about loan history, demographics, occupation, payment history, and whether or not somebody is a good customer [15]. Each instance is credit applicant. There are $n = 1,000$ instances and $d = 20$ features. The features are all either categorical or discrete. The label a indicates is a loan applicant is "good" ($y_i = 1$) or "bad" ($y_i = 0$). There are no missing values in the dataset. We renamed some of the features to be indicative of the values they represent. The dataset is self-contained and anonymous, and it includes features describing gender, age, and marital status.

| Name | Type | LB | UB | Actionability | Sign | Joint Constraints | Partition ID |
|---|---|---|---|---|---|---|---|
| Age | $\mathbb{Z}$ | 19 | 75 | No | | 0, 4, 12 | 0 |
| YearsAtResidence | $\mathbb{Z}$ | 0 | 7 | Yes | + | 0, 4, 12 | 0 |
| YearsEmployed≥1 | $\{0,1\}$ | 0 | 1 | Yes | + | 0, 4, 12 | 0 |
| CheckingAcct_exists | $\{0,1\}$ | 0 | 1 | Yes | + | 32, 33 | 30 |
| CheckingAcct≥0 | $\{0,1\}$ | 0 | 1 | Yes | + | 32, 33 | 30 |
| SavingsAcct_exists | $\{0,1\}$ | 0 | 1 | Yes | + | 34, 35 | 31 |
| SavingsAcct≥100 | $\{0,1\}$ | 0 | 1 | Yes | + | 34, 35 | 31 |
| Male | $\{0,1\}$ | 0 | 1 | No | | – | 1 |
| Single | $\{0,1\}$ | 0 | 1 | No | | – | 2 |
| ForeignWorker | $\{0,1\}$ | 0 | 1 | No | | – | 3 |
| LiablePersons | $\mathbb{Z}$ | 1 | 2 | No | | – | 4 |
| Housing=Renter | $\{0,1\}$ | 0 | 1 | No | | – | 5 |
| Housing=Owner | $\{0,1\}$ | 0 | 1 | No | | – | 6 |
| Housing=Free | $\{0,1\}$ | 0 | 1 | No | | – | 7 |
| Job=Unskilled | $\{0,1\}$ | 0 | 1 | No | | – | 8 |
| Job=Skilled | $\{0,1\}$ | 0 | 1 | No | | – | 9 |
| Job=Management | $\{0,1\}$ | 0 | 1 | No | | – | 10 |
| CreditAmt≥1000K | $\{0,1\}$ | 0 | 1 | No | | – | 11 |
| CreditAmt≥2000K | $\{0,1\}$ | 0 | 1 | No | | – | 12 |
| CreditAmt≥5000K | $\{0,1\}$ | 0 | 1 | No | | – | 13 |
| CreditAmt≥10000K | $\{0,1\}$ | 0 | 1 | No | | – | 14 |
| LoanDuration≤6 | $\{0,1\}$ | 0 | 1 | No | | – | 15 |
| LoanDuration≥12 | $\{0,1\}$ | 0 | 1 | No | | – | 16 |
| LoanDuration≥24 | $\{0,1\}$ | 0 | 1 | No | | – | 17 |
| LoanDuration≥36 | $\{0,1\}$ | 0 | 1 | No | | – | 18 |
| LoanRate | $\mathbb{Z}$ | 1 | 4 | No | | – | 19 |
| HasGuarantor | $\{0,1\}$ | 0 | 1 | Yes | + | – | 20 |
| LoanRequiredForBusiness | $\{0,1\}$ | 0 | 1 | No | | – | 21 |
| LoanRequiredForEducation | $\{0,1\}$ | 0 | 1 | No | | – | 22 |
| LoanRequiredForCar | $\{0,1\}$ | 0 | 1 | No | | – | 23 |
| LoanRequiredForHome | $\{0,1\}$ | 0 | 1 | No | | – | 24 |
| NoCreditHistory | $\{0,1\}$ | 0 | 1 | No | | – | 25 |
| HistoryOfLatePayments | $\{0,1\}$ | 0 | 1 | No | | – | 26 |
| HistoryOfDelinquency | $\{0,1\}$ | 0 | 1 | No | | – | 27 |
| HistoryOfBankInstallments | $\{0,1\}$ | 0 | 1 | Yes | + | – | 28 |
| HistoryOfStoreInstallments | $\{0,1\}$ | 0 | 1 | Yes | + | – | 29 |

**Table 9:** Separable Actionability Constraints for the processed german dataset. **Type** indicates the feature type ($\mathbb{Z}$ for integer, $\{0,1\}$ for binary). **LB**, **UB** are the lower and upper bounds for the feature. **Actionability** indicates whether the feature is globally actionable. **Sign** indicates if the feature can only increase (+) or decrease (-). **Joint Constraints** are a list non-separable constraint indices it is tied to (if any). **Partition ID** indicates which partition the feature belongs to.

**Actionability Constraints** The joint actionability constraints listed in Table 9 include:

1. DirectionalLinkage: Actions on YearsAtResidence will induce to actions on ['Age'].Each unit change in YearsAtResidence leads to:1.00-unit change in Age

2. DirectionalLinkage: Actions on YearsEmployed≥1 will induce to actions on ['Age'].Each unit change in YearsEmployed≥1 leads to:1.00-unit change in Age

3. ThermometerEncoding: Actions on [CheckingAcctexists, CheckingAcct≥0] must preserve thermometer encoding of CheckingAcct., which can only increase.Actions can only turn on higher-level dummies that are off, where CheckingAcctexists is the lowest-level dummy and CheckingAcct≥0 is the highest-level-dummy.

4. ThermometerEncoding: Actions on [SavingsAcctexists, SavingsAcct≥100] must preserve thermometer encoding of SavingsAcct., which can only increase.Actions can only turn on higher-level dummies that are off, where SavingsAcctexists is the lowest-level dummy and SavingsAcct≥100 is the highest-level-dummy.

## B.3 DETAILS FOR THE givemecredit DATASET

The givemecredit dataset is used to determine whether a loan should be given or denied [32]. The label indicates whether someone was 90 days past due in the two years following data collection. Delinquency refers to a debt with an overdue payment; this dataset is used to predict if someone will experience financial distress in the next two years. It contains information about $n = 120, 268$ loan recipients, and each instance represents a borrower. There are $d = 10$ features before preprocessing. Here, the label is $y_i = 0$ if an applicant has had a serious delinquency in two years and $y_i$ otherwise. The data is self-contained and anonymous, and it contains features describing age, income, and the number of dependents.

| Name | Type | LB | UB | Actionability | Sign | Joint Constraints | Partition ID |
|------|------|----|----|---------------|------|-------------------|--------------|
| CreditLineUtilization≥10.0 | $\{0,1\}$ | 0 | 1 | Yes | | 12, 13, 14, 15, 16 | 10 |
| CreditLineUtilization≥20.0 | $\{0,1\}$ | 0 | 1 | Yes | | 12, 13, 14, 15, 16 | 10 |
| CreditLineUtilization≥50.0 | $\{0,1\}$ | 0 | 1 | Yes | | 12, 13, 14, 15, 16 | 10 |
| CreditLineUtilization≥70.0 | $\{0,1\}$ | 0 | 1 | Yes | | 12, 13, 14, 15, 16 | 10 |
| CreditLineUtilization≥100.0 | $\{0,1\}$ | 0 | 1 | Yes | | 12, 13, 14, 15, 16 | 10 |
| MonthlyIncome≥3K | $\{0,1\}$ | 0 | 1 | Yes | + | 9, 10, 11 | 9 |
| MonthlyIncome≥5K | $\{0,1\}$ | 0 | 1 | Yes | + | 9, 10, 11 | 9 |
| MonthlyIncome≥10K | $\{0,1\}$ | 0 | 1 | Yes | + | 9, 10, 11 | 9 |
| AnyRealEstateLoans | $\{0,1\}$ | 0 | 1 | Yes | + | 17, 18 | 11 |
| MultipleRealEstateLoans | $\{0,1\}$ | 0 | 1 | Yes | + | 17, 18 | 11 |
| AnyCreditLinesAndLoans | $\{0,1\}$ | 0 | 1 | Yes | + | 19, 20 | 12 |
| MultipleCreditLinesAndLoans | $\{0,1\}$ | 0 | 1 | Yes | + | 19, 20 | 12 |
| Age≤24 | $\{0,1\}$ | 0 | 1 | No | | – | 0 |
| Age_bt_25_to_30 | $\{0,1\}$ | 0 | 1 | No | | – | 1 |
| Age_bt_30_to_59 | $\{0,1\}$ | 0 | 1 | No | | – | 2 |
| Age≥60 | $\{0,1\}$ | 0 | 1 | No | | – | 3 |
| NumberOfDependents=0 | $\{0,1\}$ | 0 | 1 | No | | – | 4 |
| NumberOfDependents=1 | $\{0,1\}$ | 0 | 1 | No | | – | 5 |
| NumberOfDependents≥2 | $\{0,1\}$ | 0 | 1 | No | | – | 6 |
| NumberOfDependents≥5 | $\{0,1\}$ | 0 | 1 | No | | – | 7 |
| DebtRatio≥1 | $\{0,1\}$ | 0 | 1 | Yes | + | – | 8 |
| HistoryOfLatePayment | $\{0,1\}$ | 0 | 1 | No | | – | 13 |
| HistoryOfDelinquency | $\{0,1\}$ | 0 | 1 | No | | – | 14 |

**Table 10:** Separable Actionability Constraints for the processed givemecredit dataset. **Type** indicates the feature type ($\mathbb{Z}$ for integer, $\{0, 1\}$ for binary). **LB**, **UB** are the lower and upper bounds for the feature. **Actionability** indicates whether the feature is globally actionable. **Sign** indicates if the feature can only increase (+) or decrease (-). **Joint Constraints** are a list non-separable constraint indices it is tied to (if any). **Partition ID** indicates which partition the feature belongs to.

**Actionability Constraints** The joint actionability constraints listed in Table 10 include:

1. ThermometerEncoding: Actions on [MonthlyIncome≥3K, MonthlyIncome≥5K, MonthlyIncome≥10K] must preserve thermometer encoding of MonthlyIncomegeq., which can only increase. Actions can only turn on higher-level dummies that are off, where MonthlyIncome≥3K is the lowest-level dummy and MonthlyIncome≥10K is the highest-level-dummy.

2. ThermometerEncoding: Actions on [CreditLineUtilization≥10.0, CreditLineUtilization≥20.0, CreditLineUtilization≥50.0, CreditLineUtilization≥70.0, CreditLineUtilization≥100.0] must preserve thermometer encoding of CreditLineUtilizationgeq., which can only decrease. Actions can only turn off higher-level dummies that are on, where CreditLineUtilization≥10.0 is the lowest-level dummy and CreditLineUtilization≥100.0 is the highest-level-dummy.

3. ThermometerEncoding: Actions on [AnyRealEstateLoans, MultipleRealEstateLoans] must preserve thermometer encoding of continuousattribute., which can only decrease. Actions can only turn off higher-level dummies that are on, where AnyRealEstateLoans is the lowest-level dummy and MultipleRealEstateLoans is the highest-level-dummy.

4. ThermometerEncoding: Actions on [AnyCreditLinesAndLoans, MultipleCreditLinesAndLoans] must preserve thermometer encoding of continuousattribute., which can only decrease. Actions can only turn off higher-level dummies that are on, where AnyCreditLinesAndLoans is the lowest-level dummy and MultipleCreditLinesAndLoans is the highest-level-dummy.

## B.4 OVERVIEW OF MODEL PERFORMANCE

We include the performance of the classifiers used in Section 4.

| | LR | | XGB | | RF | |
|---|---|---|---|---|---|---|
| Dataset | Train | Test | Train | Test | Train | Test |
| `heloc`
$n = 5,842$
$d = 43$
FICO [23] | 0.772 | 0.788 | 0.859 | 0.785 | 0.780 | 0.790 |
| `german`
$n = 1,000$
$d = 36$
Dua & Graff [15] | 0.819 | 0.760 | 0.971 | 0.794 | 0.828 | 0.766 |
| `givemecredit`
$n = 120,268$
$d = 23$
Kaggle [32] | 0.841 | 0.844 | 0.875 | 0.793 | 0.864 | 0.835 |

**Table 11:** Train and Test AUC for models across all datasets. We optimized the model's hyperparameters through randomized search and divided the data into training and testing sets at an 80% and 20% ratio.

## C SUPPLEMENTARY EXPERIMENT RESULTS

### C.1 RESPONSIVENESS OF EXPLANATIONS FOR RANDOM FORESTS

| | | All Features | | Actionable Features | | |
|---|---|---|---|---|---|---|
| Dataset | Metrics | LIME | SHAP | LIME-AW | SHAP-AW | RESP |
| `heloc`
$n = 5,842$
$d = 43$ features
$d_A = 31$ mutable
FICO [23] | % Presented with Explanations
↳ % All Features Unresponsive
↳ % At Least 1 Feature Responsive
↳ % All Features Responsive
↳ # Features Highlighted | 100.0%
86.5%
13.5%
0.0%
4.0 | 100.0%
78.2%
21.8%
0.0%
4.0 | 100.0%
77.1%
22.9%
0.0%
4.0 | 100.0%
76.7%
23.3%
0.5%
4.0 | 31.7%
0.0%
100.0%
**100.0%**
2.4 |
| `german`
$n = 1,000$
$d = 36$ features
$d_A = 9$ mutable
Dua & Graff [15] | % Presented with Explanations
↳ % All Features Unresponsive
↳ % At Least 1 Feature Responsive
↳ % All Features Responsive
↳ # Features Highlighted | 100.0%
100.0%
0.0%
0.0%
4.0 | 100.0%
89.1%
10.9%
0.0%
4.0 | 100.0%
76.6%
23.4%
0.0%
4.0 | 100.0%
64.6%
35.4%
0.0%
4.0 | 48.0%
0.0%
100.0%
**100.0%**
2.2 |
| `givemecredit`
$n = 120,268$
$d = 23$ features
$d_A = 13$ mutable
Kaggle [32] | % Presented with Explanations
↳ % All Features Unresponsive
↳ % At Least 1 Feature Responsive
↳ % All Features Responsive
↳ # Features Highlighted | 100.0%
56.5%
43.5%
0.0%
4.0 | 100.0%
26.8%
73.2%
0.5%
4.0 | 100.0%
28.4%
71.6%
1.4%
4.0 | 100.0%
21.0%
79.0%
11.4%
4.0 | 93.2%
0.0%
100.0%
**100.0%**
2.9 |

**Table 12:** Responsiveness of feature-highlighting explanations for RF for all methods and datasets. We generate explanations that highlight up to 4 top-scoring features from a given method. We report the proportion of individuals receiving an explanation (*% Presented with Explanations*) and the mean number of features in each explanation (*# Features Highlighted*). We also show the proportion of instances where all features are unresponsive (*% All Features Unresponsive*) highlighting positive values; at least one feature is responsive (*% At Least 1 Feature Responsive*), or all features are responsive (*% All Features Responsive*) highlighting the **best value**.

## C.2 FEATURE RESPONSIVENESS RANKINGS

We include a plot to show how responsive features are at different rankings by LIME, SHAP, LIME-AW, SHAP-AW and RESP for each dataset. For every denied individual, we rank features by their absolute feature importance score returned by these methods. We exclude features with 0 attribution from the rankings.

The plots below show the % of times where the feature at each rank are responsive (i.e., feature has RESP > 0). It allows us to visualize and compare how often these methods assign high attribution to responsive features.

### C.2.1 `heloc`

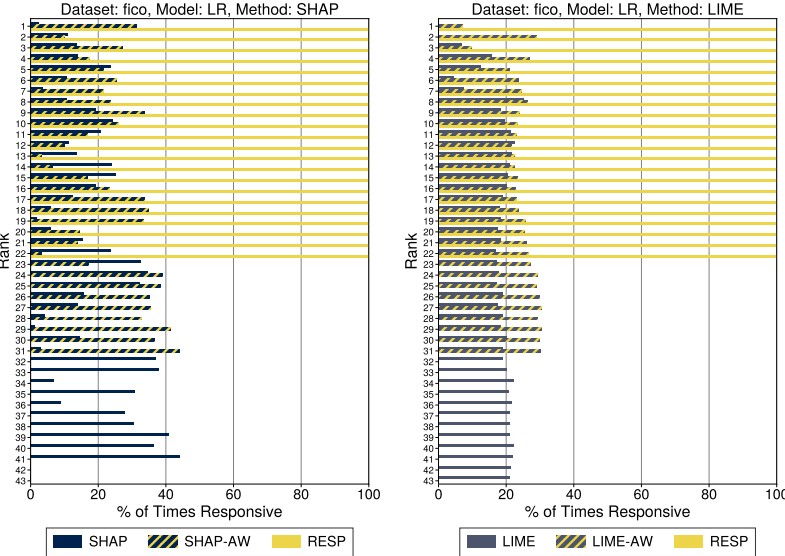

**Figure 4:** Responsiveness of features for individuals who are denied credit by the LR model on the `fico` dataset according to absolute feature attribution rank using the original feature attribution method, its action-aware variant and RESP. For each method, we report the proportion of individuals with at least one responsive intervention on a feature with the $k$-th largest score ($k$-th ranked feature). Features must have non-zero score to be included in a "rank."

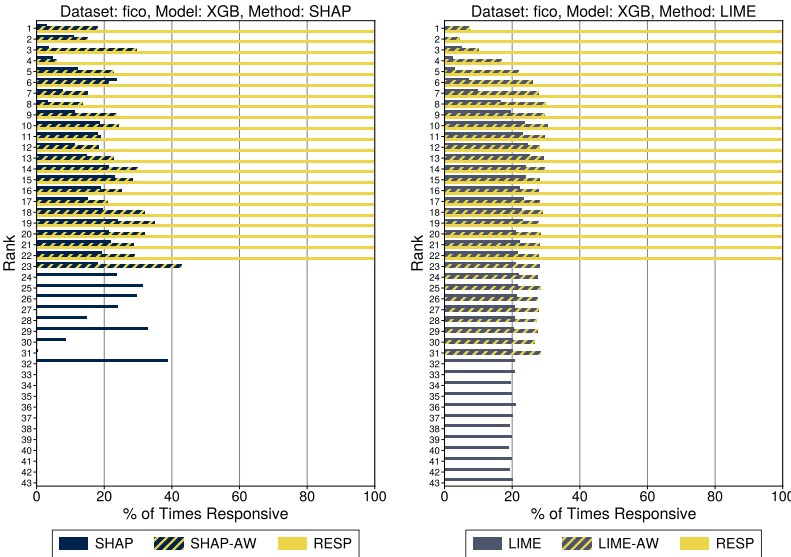

**Figure 5:** Responsiveness of features for individuals who are denied credit by the XGB model on the `fico` dataset according to absolute feature attribution rank using the original feature attribution method, its action-aware variant and RESP. For each method, we report the proportion of individuals with at least one responsive intervention on a feature with the $k$-th largest score ($k$-th ranked feature). Features must have non-zero score to be included in a "rank."

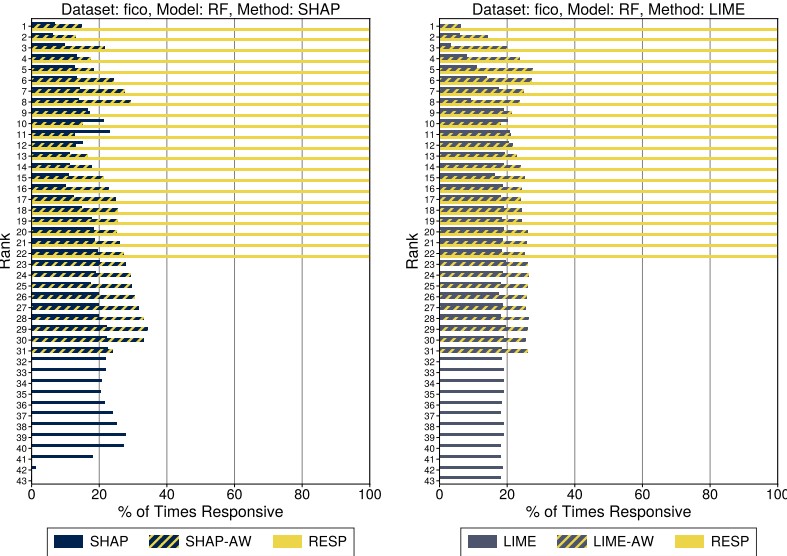

**Figure 6:** Responsiveness of features for individuals who are denied credit by the RF model on the `fico` dataset according to absolute feature attribution rank using the original feature attribution method, its action-aware variant and RESP. For each method, we report the proportion of individuals with at least one responsive intervention on a feature with the $k$-th largest score ($k$-th ranked feature). Features must have non-zero score to be included in a "rank."

### C.2.2 `german`

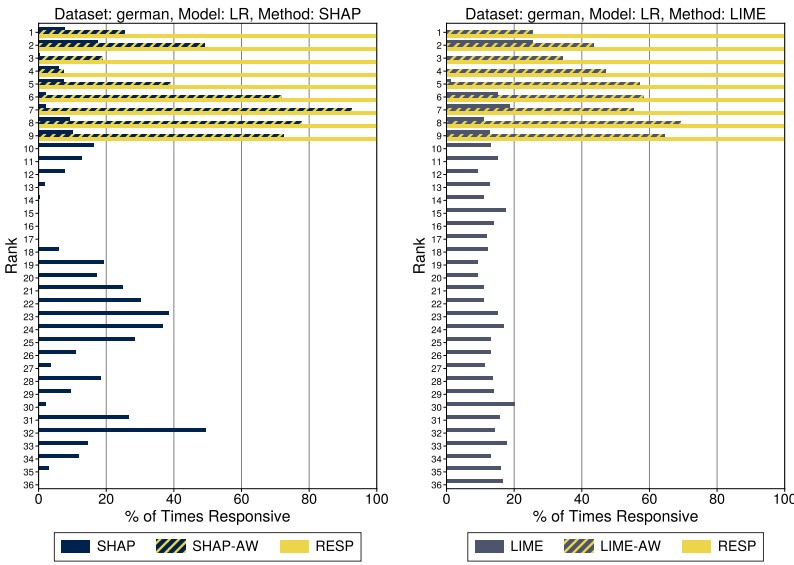

**Figure 7:** Responsiveness of features for individuals who are denied credit by the LR model on the `german` dataset according to absolute feature attribution rank using the original feature attribution method, its action-aware variant and RESP. For each method, we report the proportion of individuals with at least one responsive intervention on a feature with the $k$-th largest score ($k$-th ranked feature). Features must have non-zero score to be included in a "rank."

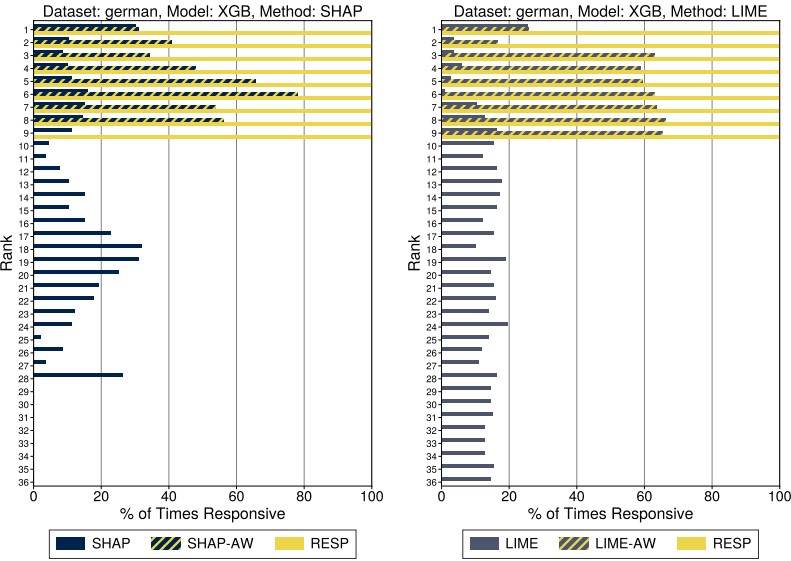

**Figure 8:** Responsiveness of features for individuals who are denied credit by the XGB model on the `german` dataset according to absolute feature attribution rank using the original feature attribution method, its action-aware variant and RESP. For each method, we report the proportion of individuals with at least one responsive intervention on a feature with the $k$-th largest score ($k$-th ranked feature). Features must have non-zero score to be included in a "rank."

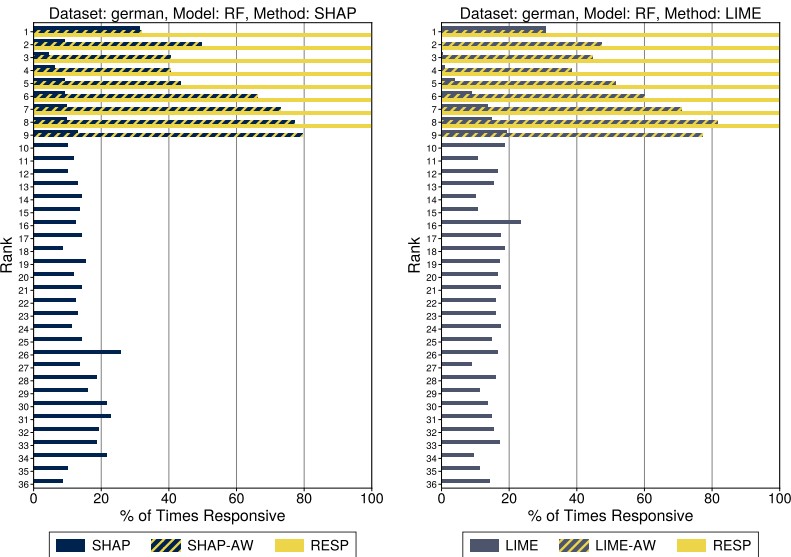

**Figure 9:** Responsiveness of features for individuals who are denied credit by the RF model on the `german` dataset according to absolute feature attribution rank using the original feature attribution method, its action-aware variant and RESP. For each method, we report the proportion of individuals with at least one responsive intervention on a feature with the $k$-th largest score ($k$-th ranked feature). Features must have non-zero score to be included in a "rank."

### C.2.3 `givemecredit`

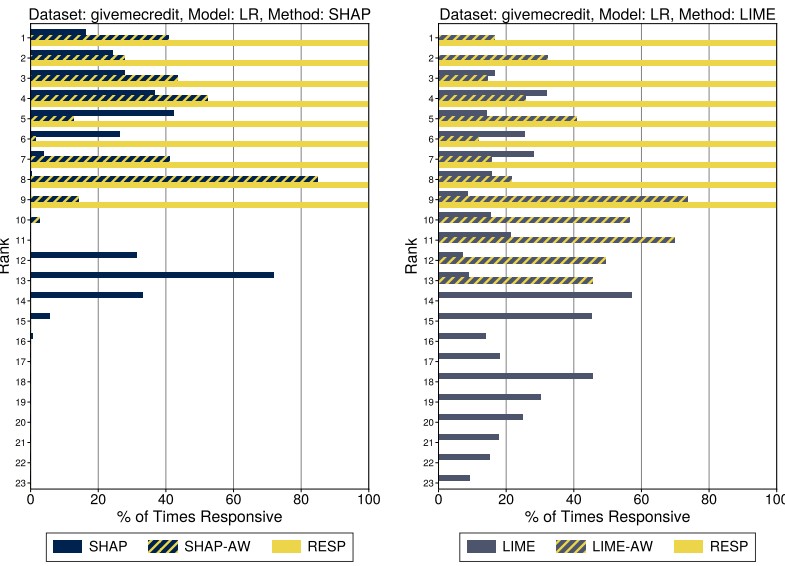

**Figure 10:** Responsiveness of features for individuals who are denied credit by the LR model on the `givemecredit` dataset according to absolute feature attribution rank using the original feature attribution method, its action-aware variant and RESP. For each method, we report the proportion of individuals with at least one responsive intervention on a feature with the $k$-th largest score ($k$-th ranked feature). Features must have non-zero score to be included in a "rank."

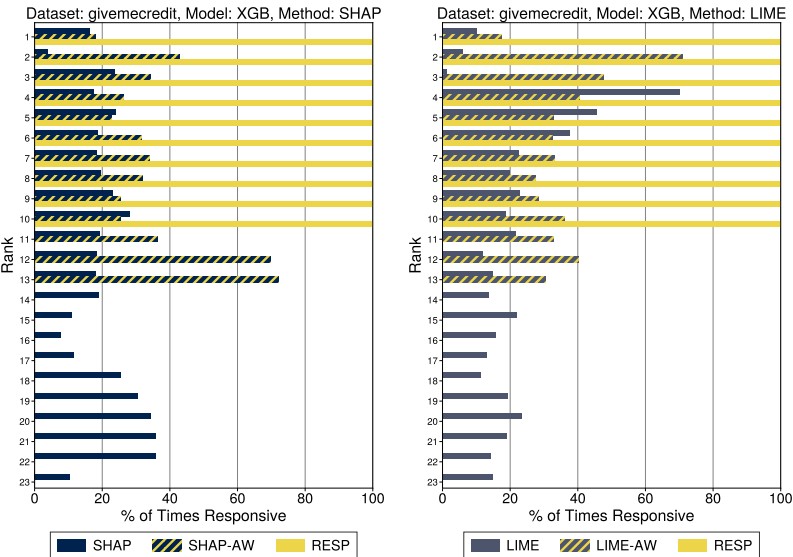

**Figure 11:** Responsiveness of features for individuals who are denied credit by the XGB model on the `givemecredit` dataset according to absolute feature attribution rank using the original feature attribution method, its action-aware variant and RESP. For each method, we report the proportion of individuals with at least one responsive intervention on a feature with the $k$-th largest score ($k$-th ranked feature). Features must have non-zero score to be included in a "rank."

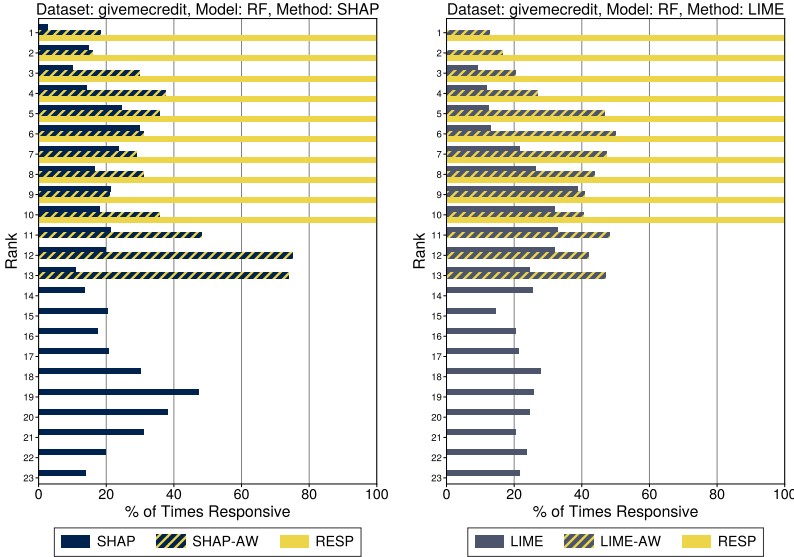

**Figure 12:** Responsiveness of features for individuals who are denied credit by the RF model on the `givemecredit` dataset according to absolute feature attribution rank using the original feature attribution method, its action-aware variant and RESP. For each method, we report the proportion of individuals with at least one responsive intervention on a feature with the $k$-th largest score ($k$-th ranked feature). Features must have non-zero score to be included in a "rank."

# D  SUPPLEMENTARY CASE STUDY DETAILS

## D.1  ACTIONABILITY CONSTRAINTS

The joint actionability constraints listed in Table 13 include:

| Name | Type | LB | UB | Actionability | Sign | Joint Constraints | Partition ID |
|---|---|---|---|---|---|---|---|
| Age | $\mathbb{Z}$ | 21 | 103 | No | | 0, 8, 10 | 0 |
| HistoryOfLatePaymentInPast2Years | $\{0,1\}$ | 0 | 1 | Yes | + | 0, 8, 10 | 0 |
| HistoryOfDelinquencyInPast2Years | $\{0,1\}$ | 0 | 1 | Yes | + | 0, 8, 10 | 0 |
| NumberRealEstateLoansOrLines | $\mathbb{Z}$ | 0 | 100 | Yes | + | 5, 6 | 5 |
| NumberOfOpenCreditLinesAndLoans | $\mathbb{Z}$ | 0 | 100 | Yes | + | 5, 6 | 5 |
| NumberOfDependents | $\mathbb{Z}$ | 0 | 20 | No | | – | 1 |
| DebtRatio | $\mathbb{R}$ | 0.0 | 61106.5 | Yes | | – | 2 |
| MonthlyIncome | $\mathbb{Z}$ | 0 | 3008750 | Yes | | – | 3 |
| CreditLineUtilization | $\mathbb{R}$ | 0.0 | 50708.0 | Yes | | – | 4 |
| HistoryOfLatePayment | $\{0,1\}$ | 0 | 1 | No | | – | 6 |
| HistoryOfDelinquency | $\{0,1\}$ | 0 | 1 | No | | – | 7 |

**Table 13:** Separable Actionability Constraints for the processed continuous `givemecredit` dataset. **Type** indicates the feature type ($\mathbb{Z}$ for integer, $\{0,1\}$ for binary). **LB**, **UB** are the lower and upper bounds for the feature. **Actionability** indicates whether the feature is globally actionable. **Sign** indicates if the feature can only increase (+) or decrease (-). **Joint Constraints** are a list non-separable constraint indices it is tied to (if any). **Partition ID** indicates which partition the feature belongs to.

1. DirectionalLinkage: Actions on `NumberRealEstateLoansOrLines` will induce to actions on ['NumberOfOpenCreditLinesAndLoans'].Each unit change in `NumberRealEstateLoansOrLines` leads to:1.00-unit change in `NumberOfOpenCreditLinesAndLoans`

2. DirectionalLinkage: Actions on `HistoryOfLatePaymentInPast2Years` will induce to actions on ['Age'].Each unit change in `HistoryOfLatePaymentInPast2Years` leads to:2.00-unit change in `Age`

3. DirectionalLinkage: Actions on `HistoryOfDelinquencyInPast2Years` will induce to actions on ['Age'].Each unit change in `HistoryOfDelinquencyInPast2Years` leads to:2.00-unit change in `Age`

## D.2 MODEL PERFORMANCE

| | XGB | |
|---|---|---|
| Dataset | Train | Test |
| `givemecredit` $n = 120, 268$ $d = 11$ Kaggle [32] | 0.937 | 0.830 |

**Table 14:** Model Performance of XGB model on the `givemecredit` dataset for Section 5

