# OpenReview forum: "Feature Responsiveness Scores: Model-Agnostic Explanations for Recourse"
_ICLR.cc/2025/Conference — ICLR 2025 Poster_

### Official Review · Reviewer_pPyj · 2024-11-01

**Soundness:** 3
**Presentation:** 3
**Contribution:** 2
**Rating:** 8
**Confidence:** 3

**Summary:**

The authors propose feature responsiveness scores to highlight the features that can be changed to receive a better model outcome and assess how feature attribution methods can inflict harm by providing users with reasons without recourse.

**Strengths:**

The authors propose a novel way of looking at actionability.
The authors perform comprehensive experiments using three datasets and explainability methods and avail their code.

**Weaknesses:**

**Weaknesses and questions**

- There are endogenous and exogenous actionable features. When defining the actionability constraints (e.g., from the way authors define them - table 2 for example), it’s non-trivial to fully capture the relationship between features, especially in a way that reflects varied real-world scenarios. The action set and reachability set might not accurately capture these dynamics. For example, by making age immutable, a user is limited to recourse that doesn’t involve age changing (e.g. figure 2), and in some settings, e.g., education and age, if an individual changes from education:1 to education:3, age changes as well. As a result, the formulation of the responsiveness score might be limited.
- While some features are not actionable, they could be highly predictive and ethical. For example in predicting who is admitted to a K1 class, age is a very predictive feature despite being a sensitive attribute and not being actionable. Similarly, disability is a very predictive feature in predicting eligibility for the Paralympics despite being a sensitive attribute and not being actionable. So although the feature might not be actionable and not eligible for actionable recourse recommendations, if methods like LIME and SHAP attach a high feature attribution score to it, it’s not a negative thing. I think the responsiveness score is context-dependent and the proposed formulation could be stronger with some way of filtering instances where the score is most informative or applicable.
- The reliance on the computation of the responsive score on the reachable set is very limiting. This is because 1) The decision-maker has to define the reachability set based on the action set (which might be limited by the training data manifold and is susceptible to breaking causality and the relationship between features). 2) It’s hard to compute the responsiveness score in cases where features are continuous and sampling might introduce bias that's propagated and exacerbated in the explanation. 3) For methods like LIME or SHAP where the objective is to get a sense of the most important features among a set of features or determine the value of features to the model, defining an action set or individualized recourse might be out of scope.
- The formulation of actionability and subsequently responsiveness score such that the number of features changed = number of actions taken is rarely the case in real-world settings. For example, while a student might be required to get at least a grade A in class w and at least a grade B in class z, one action might lead to the achievement of both grades (features).
- In addition to the quantitative results (tables and numbers ), it might be helpful to add more qualitative results as well. For example, on page 9, lines 467 to 468, the authors briefly do this.

**Minor  or other observations**

- I think it might be interesting to further investigate the responsiveness score at an individual or instance level. I imagine that in a real-world setting, the responsiveness score might be limited by what actions different individuals have access to, varied user preferences and strengths/circumstances (potentially captured by individual costs), and so on.
- Additionally, in several recourse methods, several alternative sets of actions, e.g., with different costs, or diverse actionable features, are returned, where e.g., in some instances, the least costly is chosen. \
How well or easily is the responsiveness score scalable to this setting? Additionally, how do authors determine sufficient alternatives recourse as cutoff in that case, for example would it be dependent on diversity of the features recommended in recommended varied recourse, or number of alternative recourse, etc?
- Given the dependence on a manual action set definition, the responsiveness score is susceptible to biases and replication or reproducibility challenges.
- The figures and tables communicate what the authors want to convey. However, the text font is very small. Authors could find ways to increase the font to improve the readability of the figures and tables.
- Given that only users that have received a negative outcome get recourse, would it be computationally cheaper to more generally bound the reachability set, since the action set is more general -specific to the training set-?
- How does a change in training data distribution affect the feature responsiveness score?
- The reachability set and responsiveness scores are more likely to leak significant information about the model than the local explanation tools like SHAP and LIME.

**Questions:**

The questions are integrated with the context in the weakness section. Authors should please respond to the questions added in the sub-section "weaknesses and questions"

---

> ### Author Response · Authors · 2024-11-22
>
> Thank you for your time and feedback!
>
> > by making age immutable, a user is limited to recourse that doesn’t involve age changing (e.g. figure 2), and in some settings, e.g., education and age, if an individual changes from education:1 to education:3, age changes as well.
>
> We agree that it is important to capture these constraints – and we do! The actionability constraints we use in our experiments allow us to account for settings where changing an actionable feature can induce changes in a non-actionable feature. In our experiments, we use this constraint to ensure that changing `years_in_residence` leads to changes in `age`. We can apply something similar to capture the relationship between education and age.
>
> > So although the feature might not be actionable and not eligible for actionable recourse recommendations, if methods like LIME and SHAP attach a high feature attribution score to it, it’s not a negative thing.
>
> Thanks for bringing this up!
>
> We agree that LIME and SHAP would be correct to assign high scores in the example you pointed out.
>
> We want to be clear that the "negative effects" that are discussed are not failures in attribution or algorithm design, but rather in the methods that we choose for a given application. Maybe the best way to put this is that – before we can determine whether the output of LIME and SHAP is good or bad – we have to answer the following questions:
> 1. Is this an application where individuals have a right to recourse?
> 2. Is this an application where we want to provide individuals with explanations that lead to recourse?
>
> In the driving application for the paper – i.e., explaining adverse credit decisions – the answer to both of these questions is "yes." In this case, we want to explain the predictions of a model to provide individuals with recourse (in part), and there is a right to recourse (see e.g., [Taylor 1980](​​https://digitalcommons.law.buffalo.edu/buffalolawreview/vol29/iss1/4/) for the history of the Equal Credit Opportunity Act in the US); if not, we would be cutting off access to credit.
>
> In this setting, LIME and SHAP have negative effects because they highlight features that are not aligned with these broader goals. On the one hand, our explanations may include features that are "important" but non-actionable (which would not help consumers achieve recourse). Alternatively, we could end up with an explanation where every feature is actionable but not-responsive (which would harm consumers).
>
> Stepping back, there are broader applications where we can use responsiveness scores – even though there is no right to recourse, or the explanations are not designed for consumers. These include:
> - Organ transplant: We can use responsiveness scores to stress test organ transplant decision models. The goal would be to increase transparency in these models and evaluate their fairness (see [this article](https://www.ft.com/content/5125c83a-b82b-40c5-8b35-99579e087951) for an example of why a model could be unfair).
> - Recidivism prediction: In a similar fashion, we can evaluate the fairness of risk scoring models. Take [this](https://anonymous.4open.science/r/responsiveness-iclr-2025/figures/OGS.pdf) simple scoring model for example. Although these features are not mutable (or actionable), we can assess the model’s fairness by looking at the responsiveness of the subject’s attributes.
>
> Please let us know if you agree. We can include a discussion on this point in our concluding remarks.
>
> >  The reliance on the computation of the responsive score on the reachable set is very limiting [...] 1) The decision-maker has to define the reachability set based on the action set (which might be limited by the training data manifold and is susceptible to breaking causality and the relationship between features)
>
> We think that there might be a misunderstanding. The decision-maker only has to define the action set (see paragraph “Accounting for Actionability” in Section 2). We can then construct the reachable set using enumeration or via sampling. The computation associated with these tasks is not all that bad. On the `givemecredit` dataset ($n = 120,268$ points and $d = 23$ features), enumerating the reachable set for each instance takes less than an hour (c.f., < 1 minute for sampling). In all cases, the reachable sets can be re-used for other tasks (e.g., as we update the model).

---

> ### Author Response · Authors · 2024-11-22
>
> >  The formulation of actionability and subsequently responsiveness score such that the number of features changed = number of actions taken is rarely the case in real-world settings.
>
> We think there might be a misunderstanding here as well! We are not entirely sure what you meant, but we do not assume that " the number of features changed = number of actions taken”. To be clear, the responsiveness score quantifies the availability of recourse along a single-feature axis. Figure 2 and this [illustration](https://anonymous.4open.science/r/responsiveness-iclr-2025/figures/counterfactual_vs_resp.png) should help illustrate what we mean. There are a total of 9 actions along each feature axis, 4 of them lead to recourse for $x_1$ and 1 for $x_2$ (green area). Please let us know if this clears things up!
>
> > [...] in several recourse methods, several alternative sets of actions, e.g., with different costs, or diverse actionable features, are returned, where e.g., in some instances, the least costly is chosen.
>
> > How well or easily is the responsiveness score scalable to this setting? [...]
>
> Thanks for bringing this up! We can definitely extend our original responsiveness construct to take a cost function into account, so that we weight actions according to their cost. This "weighted" variant is easy to implement: given a cost function, we can compute the cost of each reachable point and return a feature’s cost-weighted responsiveness (some variant of multiplying the cost by responsiveness of the action). We elected not to include the weighted version because it may invalidate the natural probabilistic interpretation. In practice, it is challenging to specify a cost function: costs will vary from person to person and should be elicited from individuals.
>
> > Given the dependence on a manual action set definition, the responsiveness score is susceptible to biases and replication or reproducibility challenges.
>
> It is true that the action set must change across tasks. However, practitioners are not free to specify the action set to their taste. This is because for each dataset with semantically meaningful features, there are what we call **inherent constraints.** These are constraints that stem from immutability (e.g., sex or other protected attributes), physical/biological limits (e.g., age can only increase) and feature encoding (e.g., you can only get more education). These are indisputable constraints that every practitioner would have to include and can do so with ease based on the data dictionary.
>
> Moreover, we are well-equipped to deal with replication and reproducibility challenges. The constraints in the action set can be written in natural language and be included alongside a data dictionary. As a result, there are no privacy concerns in sharing this, since it does not necessarily contain the data itself. In fact, the action set can be shared as a [Python script](https://anonymous.4open.science/r/responsiveness-iclr-2025/scripts/experiment/setup_dataset_actionset_fico.py) or as a Python object. For instance, the action set for our largest dataset, `heloc`, is 13 KB.
>
> > How does a change in training data distribution affect the feature responsiveness score?
>
> We were not quite sure about what you meant here. In practice, the answer depends on whether you would train a new model after the distribution changes, or not. We'll answer both cases below.
>
> Under a fixed model $h$, feature responsiveness scores do not change. Recall the formula for the responsiveness scores:
> $$\mu_j = \frac{1}{|R_j|} \sum_{x \in R_j} \mathbb{1}[h(x) = 1]$$
> We observe that none of the components depend on the distribution of the training data.
>
> If you allow the model to change, then the feature responsiveness scores would change. In this case, the change would stem from changes in the decision boundary. However, if we have the (enumerated or sampled) reachable set, the score calculation process under the new model is trivial.
>
> > The reachability set and responsiveness scores are more likely to leak significant information about the model than the local explanation tools like SHAP and LIME.
>
> We agree that there is some potential to leak information, but do not see how approach is more prone to leakage compared to existing approaches methods like LIME or SHAP. In comparison to LIME, for example, the responsiveness score may not provide meaningful information about the local decision boundary. In comparison to SHAP, it does not provide information on the training dataset. In comparison to existing approaches, this approach also provides some mechanism to control what we report without violating the recourse guarantee (e.g., we can choose not to report certain features and still provide a meaningful guarantee of recourse). We recognize that this is speculative and merits further investigation. We will list it as a potential limitation.

---

> ### Author Response · Authors · 2024-11-22
>
> > Authors could find ways to increase the font to improve the readability of the figures and tables.
>
> These are now updated! Please let us know if it reads better now.

---

> > ### Author Response · Authors · 2024-11-26
> > **Checking In**
> >
> > We just wanted to check in and see if you’ve had a chance to read our comments! We have a few more days left in the discussion period, so if you have additional questions or concerns, let us know, and we can do our best to address them over the coming days. Alternatively, if we have addressed your concerns, please consider raising your score!

---

> ### Comment · Reviewer_pPyj · 2024-11-27
>
> Thank you, authors, for addressing my questions.
>
> I agree with the authors that two considerations (Is this an application where individuals have a right to recourse? Is this an application where we want to provide individuals with explanations that lead to recourse?) are crucial. However, another efficient consideration is stakeholders, i.e., "Who is this explanation for?"
> In my opinion, it is acceptable for explanations to focus on explaining the prediction without necessarily being actionable and others to focus on actionability - translating potential changes into meaningful actions. For instance, explanations designed for auditors or decision-makers can provide valuable insights that help improve the model, even if they do not directly support recourse. In my view, it is okay to separate these things.
> In addition, I do not entirely agree with the suggestion that recourse is unnecessary in the context of recidivism prediction. On the contrary, I believe recourse is critical in such scenarios.
>
> While some author responses were satisfactory and some weren't (probably personal opinions, e.g., on action sets and the example above), I appreciate the authors' efforts and believe the paper makes meaningful contributions to the community. I will raise my score to 8.

---

### Official Review · Reviewer_6js1 · 2024-11-05

**Soundness:** 2
**Presentation:** 2
**Contribution:** 2
**Rating:** 6
**Confidence:** 4

**Summary:**

This work studies feature attribution and its relationship to recourse. It demonstrates that many feature attribution techniques do not highlight features that can lead to recourse, i.e. these methods do not identify features that if an individual were to change them, the model. prediction would be different. This work then addresses this issue by proposing a way to score features on the basis of responsiveness – i.e., the probability that an individual can attain a desired outcome by changing a specific feature. The paper presents efficient methods to compute such scores and demonstrates via experiments that their proposes responsiveness scores highlight features that can lead to recourse.

**Strengths:**

1) I believe this paper aims to address a very important problem with current feature attribution methods, that is, the features these methods identify as important are rarely those that can be modified/changed so that a different prediction may occur.

2) I believe the idea of action and reachable sets is an interesting one and I appreciate the authors trying to make these notions precise via theory.

3. The paper performs numerous experiments to demonstrate their result

**Weaknesses:**

1) I don't particularly think this work is very novel. There has been numerous works that propose counterfactual feature attribution methods, i.e. those that identify important features as the ones that, when changed, lead to a different prediction. Are these not pretty much responsiveness scores in the language of this paper. I advise the authors to take a look at these papers and address how their proposed notion of responsiveness is different than the notions proposes in the following works

https://proceedings.mlr.press/v162/dutta22a/dutta22a.pdf
https://arxiv.org/abs/2409.20427
https://arxiv.org/pdf/1912.03277

The beginning of the 3rd page of the first paper [Dutta et al 2022] in fact list many works that focus on develop feature importance scores that identify important features as those when changed, lead to a different prediction

2) How are the actionable sets defined exactly? To be correct the only way to properly understand model behavior is to see how its predictions change while modifying features such that the new feature vector x' still comes from the same distribution that the original feature vector comes from. This is because the model was trained on samples from this distribution and so to evaluate how it performs on samples that may not make sense does not make sense to me. Is the reachable set trying to capture all "feasible" ways that we can change the feature vector? How do identify this reachable set?

3) Overall I have concerns on the novelty and limited impact of this work. These ideas have been explored in numerous works and whats outlined in the paper seems to only apply to simple settings with discrete features where its clear what the reachable set is.

**Questions:**

What is line 96 saying?
- "By highlighting features that can be changed to consumers who could not attain a different outcome
- I thought current methods fail to identify features that are actionable, i.e. those that could be changed to achieve a different, hopefully desirable outcome? I am not sure what this line is saying.

---

> ### Author Response · Authors · 2024-11-22
>
> Thank you for your time and feedback! We believe there might be some misunderstandings and would like to help clear them up.
>
> > Numerous works that propose counterfactual feature attribution methods, [...] Are these not pretty much responsiveness scores in the language of this paper. I advise the authors to take a look at these papers and address how their proposed notion of responsiveness is different [...]
>
> There seems to be a major misunderstanding that we would like to address. To be clear, we are very familiar with these papers. There is a fundamental difference between what these methods return and what we are trying to capture using responsiveness scores.
>
> Given a model $h: \mathcal{X} \to \mathcal{Y}$ and a point $\mathbf{x}_0$:
> - These other methods return a single action that flips the prediction of the model, i.e., they return $a \in \mathbb{R}^d$ such that $h(\mathbf{x}_0 + \mathbf{a}) = 1$
> - In contrast, responsiveness scores measure the *proportion* of actions that lead to recourse in each dimension of the feature space, i.e., for each $j \in [d]$, we return $\mu_j(\mathbf{x}_0) = P(h(\mathbf{x}_0 + \mathbf{a} = 1 | \mathbf{a} \text{ is a single feature action on } j)$.
>
> As a motivating example, consider a task where we applied both methods to return a 1D action. (see this [figure](https://anonymous.4open.science/r/responsiveness-iclr-2025/figures/counterfactual_vs_resp.png). There are a total of 9 actions along each feature axis, 4 of them lead to recourse for $x_1$ and 1 for $x_2$ (green area). In this case, the method you are referencing would return the feature with the closest recourse action ($x_1$). In contrast, we would highlight the feature that admits the most actions ($x_2$). The latter is more valuable in settings where we are highlighting features since it maximizes the chances of recourse.
>
> The example above is only meant to highlight a fundamental conceptual difference. In general, our approach has several other important differences:
> - *Versatility*: our approach is model-agnostic (c.f,, [Dutta et al.](https://arxiv.org/abs/2207.02739))
> - *Actionability*: our approach can enforce complex deterministic actionability constraints (c.f,, [Dutta et al.](https://arxiv.org/abs/2207.02739), [Mahajan et al.](https://arxiv.org/pdf/1912.03277)), without having to learn causal models.
> Stepping back, we realize that this should have been clearly stated in the related work section, and would benefit from a diagram like the one above. We will update the manuscript accordingly.
>
> > Whats outlined in the paper seems to only apply to simple settings with discrete features where its clear what the reachable set is.
>
> This also seems to be a misunderstanding! What's covered in the paper applies to all features (discrete or continuous features for any classifier). The reachable set is not "hand crafted" but defined in a principled way as described below (see also Section 3.2).
>
> > How are the actionable sets defined exactly? [...]
>
> We define the action set as a collection of actionability constraints (like the ones in Table 2). We then construct it as follows:
> 1. We elicit constraints on each feature and from practitioners or from decision subjects.
> 2. We convert each of these conditions into mathematical constraints (see Table 2).
>
> In practice, this process would be carried out by practitioners or could be elicited from decision subjects, which could lead to changes. However, every dataset admits a basic set of indisputable constraints (e.g., everyone will agree that one’s sex and age are not actionable).
>
> In our experiments, we use this minimal set of constraints. You can see them in Appendix A.
>
> > Is the reachable set trying to capture all "feasible" ways that we can change the feature vector? How do I identify this reachable set?
>
> We construct the reachable set as described in Section 3.2. Given a feature, we define an action set for feature $j$, $A_j$. If $A_j$ is discrete, we enumerate; if $A_j$ is continuous, we sample. The reachable set for $j$ is entirely defined by the action set. This is an equivalent way to represent the same information.
>
> > What is line 96 saying?
>
> We are describing the problem of *reasons without recourse*. This happens when we give explanations to individuals who are assigned a fixed prediction (i.e., they cannot change their prediction under actionability constraints). We’ve fixed this in our updated manuscript. Please let us know if this helps.

---

> > ### Author Response · Authors · 2024-11-26
> > **Checking In**
> >
> > We just wanted to check in and see if you’ve had a chance to read our comments! We have a few more days left in the discussion period, so if you have additional questions or concerns, let us know, and we can do our best to address them over the coming days. Alternatively, if we have addressed your concerns, please consider raising your score!

---

> > > ### Comment · Reviewer_6js1 · 2024-11-26
> > > **Response to authors**
> > >
> > > Thanks for the responses. My misunderstanding of what responsiveness scores are has been cleared up so thank you! For that I have raised my score.
> > >
> > > I still do have some lingering questions about the actionable sets.
> > >
> > > 1) How does one **define** an actionable set. As I understand, A_j(x) is just all the feasible ways I can change x_j correct? Is this not just all the possible x_j for which P(x_j | X_{-j} = x_{-j}) > 0? If so, this is not necessarily easy to define.
> > >
> > > 2) Does your framework allow for multidimensional actions? Right now you can define responsiveness score of a single feature j but I can imagine that this may = 0. However, if I add another feature k, the responsiveness score of (j,k) might be very high?

---

> > > > ### Author Response · Authors · 2024-11-27
> > > >
> > > > Thank you for your quick response! We would be more than happy to answer your questions:
> > > >
> > > > > 1. How does one **define** an actionable set. As I understand, A_j(x) is just all the feasible ways I can change x_j correct? [...]
> > > >
> > > > Yes, you are correct that $A_j(\mathbf{x})$ is a set that contains all feasible “actions” on feature $j$ for $\mathbf{x}$!
> > > > As we’ve mentioned in our response above, we do this by specifying **actionability constraints.** For example, if we have an ordinal encoding of a feature for `education` (0 for no degree, 1 for high school diploma, … 5 for doctorate degree, etc.), we might want to enforce a monotonicity constraint — i.e., you can’t go down the ordinal scale. Given a point with $x_j = 3$, our constraints immediately tell us that $A_j(\mathbf{x}) = \\{4,5\\}$. Note that every feature has at least one constraint: bounds. Our setup in Section 2 assumes that all semantically meaningful features are bounded. So without any other constraints on feature $k$, $A_k(\mathbf{x}) = [l_k, u_k]$ (or an equivalent set if $k$ is a discrete feature), where $l_k$ and $u_k$ are the lower and upper bounds of feature $k$.
> > > >
> > > > But the important takeaway here is that *actionability constraints* **define** $A_j(\mathbf{x})$. This is why we store action sets as a list of constraints. Then we use the list of constraints to build the reachable set to calculate responsiveness scores, either by sampling or enumeration (Section 3.3).
> > > >
> > > > > 2. Does your framework allow for multidimensional actions? [...]
> > > >
> > > > In principle, yes! We can easily adapt our algorithms in Section 3.3 to accommodate multidimensional actions. In fact, we do this in our experiments to verify recourse and identify individuals with fixed predictions. The `n-D Rec` metric in **Table 3** indicates the proportion of denied individuals who can only achieve recourse through multidimensional (n-D) actions.
> > > >
> > > > This is part of the motivation for our paper. Current regulations favor 1D actions due to user understandability of explanations involving n-D actions. It is difficult to portray multi-feature changes because feature-highlighting explanations, like adverse action notices (in consumer finance), are lists of features that do not characterize feature dependencies or interactions. Simply put, consumers who see them are likely to believe that they can be changed independently. However, our results highlight how this can be restrictive and how we can sometimes address this by suggesting multidimensional actions.
> > > >
> > > >
> > > > Please let us know if this clears things up or if you have further questions!

---

> > > > > ### Comment · Reviewer_6js1 · 2024-11-27
> > > > > **Response to authors**
> > > > >
> > > > > Thank you for the response and I appreciate the authors for engaging with me. I understand that you specify actionability constraints define $A_j(x)$, and I appreciate the example.
> > > > >
> > > > > My follow-up question is, do these actionability constraints take into other features? Take your example with education and suppose we have an individual who has obtained a bachelor's degree, and we impose a monotonicity constraint. Thus, this person can only get a master's/PhD/higher ed degree. However, suppose another feature is "savings," and suppose this individual currently does not have much savings. Then I imagine the constraints on education **could** become more restrictive, i.e., a master's program may be infeasible due to the price, or a PhD may be infeasible because the person cannot afford to stop working/making a salary. How does your framework handle this scenario? Do the actionability constraints reflect this complex constraint?
> > > > >
> > > > > Also, how do you know what the "right" actionability constraints are? Are they informed by domain experts?

---

> > > > > > ### Author Response · Authors · 2024-11-27
> > > > > >
> > > > > > Thanks for the follow up questions!
> > > > > > > My follow-up question is, do these actionability constraints take into other features? [...] How does your framework handle this scenario? Do the actionability constraints reflect this complex constraint?
> > > > > >
> > > > > > We aren't quite sure if you mean whether we can handle it or whether we should handle it. In terms of capability, we could handle something like this. We would add a constraint like: to increase education from 3 (BS) to 4 (MS), savings must decrease by 50K (see “causal implication” in Table 2). In terms of whether we should handle it, the answer is less clear.
> > > > > >
> > > > > >
> > > > > > Knowing nothing else, we would not; a constraint like above is a strong assumption. There may be a way to obtain an MS degree without cutting into savings (e.g., scholarship, RA/TAship). In this case, there is a possibility we "miss" fixed predictions that occur due to individual constraints of not being able to attain a scholarship. On the flip side, every fixed prediction we flag would be one that is based on solid, indisputable constraints.
> > > > > >
> > > > > >
> > > > > > In practice, we would implement them if either a domain expert suggested such constraints or they were elicited from agents.
> > > > > >
> > > > > >
> > > > > > > Also, how do you know what the "right" actionability constraints are? Are they informed by domain experts?
> > > > > >
> > > > > >
> > > > > > Thanks for bringing up a very important question!
> > > > > >
> > > > > >
> > > > > > The "right" constraints are, quite frankly, hard to know a priori.
> > > > > >
> > > > > >
> > > > > > A sensible approach could be to start with a minimal set of constraints that everyone agrees on (what we call **indisputable** constraints). Then we could add more constraints from domain experts or potentially customers.
> > > > > >
> > > > > >
> > > > > > In theory, we can have domain experts state each condition one by one in a list as natural language and share it among themselves. If multiple experts disagree, they can deliberate on them. In such cases, we can potentially simplify the deliberation by discarding constraints that do not matter via an ablation study. In the worst case, we can choose their common constraints.
> > > > > >
> > > > > >
> > > > > > In practice, we have the capability to handle it, but we have not done this extensively (the `HELOC` dataset was put together with an expert in the credit scoring industry). We expect that this kind of elicitation work will be more complicated than it seems — experts will disagree, and there will be ambiguity (as in the education case above).
> > > > > >
> > > > > >
> > > > > > We imagine that we would encompass all using a conservative approach—where we only implement **indisputable** actionability constraints. In this case, we would flag when we are sure that a prediction is fixed (e.g., the preclusion stems from actionability constraints that are not subject to debate or disagreement). The intuition here is that recourse availability decreases with more (and stronger) constraints. If an individual has a fixed prediction under lax constraints, we are sure that they do not have recourse under stronger constraints.
> > > > > >
> > > > > >
> > > > > > The downside of this regime is that we might highlight features that overestimate responsiveness because they view some changes as "actionable" even when they are not. This could lead to harm—albeit it is more of a gray area because the actionability constraints that were violated were disputable.
> > > > > >
> > > > > >
> > > > > > At the same time, it may not matter—e.g., say that we present a consumer with four features, two of which violate some actionability constraints that we did not consider (e.g., savings balance in your example). In this case, they may realize that one of these is not viable and focus on the other two changes. Part of giving a consumer multiple features that can be changed independently is because this provides them with flexibility/buffer to account for their private restrictions. After accounting for their private constraints, there may still be some path to recourse.

---

> > > > > > > ### Comment · Reviewer_6js1 · 2024-11-27
> > > > > > > **Response**
> > > > > > >
> > > > > > > thank you for the thorough answers. I am raising my score and believe the paper is a nice contribution!

---

### Official Review · Reviewer_PaRQ · 2024-11-08

**Soundness:** 4
**Presentation:** 4
**Contribution:** 4
**Rating:** 8
**Confidence:** 5

**Summary:**

The paper proposes a new concept within interpretability of ML models, focusing on explanations for recourse. Even though the approach is specially proposed, developed and discussed within the context of lending, I believe it has general applicability for a broader set of problems. Basically, instead of considering shapley values or other concepts to assign a contribution/importance to some input feature, it concentrates instead on assessing sensitivities of the forecasts to changes in the features. These features that could be changed are seen as recourse. While the idea is very simple, and could be seen as a simple sensitivity measure, I believe it can be a very understandable and powerful approach to convey information that would support interpretability much better than Shapley values, for many applications.

**Strengths:**

I clearly agree with the starting point of the paper, aiming to focus on another view of interpretability, where for a number of applications, only features that are actionable are relevant, while one would be interested in seeing how different outcomes may be if values for these features would change (even slightly). It is crucial to go away from thinking that Shapley values (and other similar approaches) are the go-to approach to bring interpretability to ML. Here, the approach is described in a rigorous and pedagogical manner. The concepts and main measure proposed (in Definition 4) are simple yet powerful. The methodological contribution is sound, original and valuable. Personally, I enjoyed reading the paper and it made me think about a lot of potential use cases and extensions.

**Weaknesses:**

In my view the only weakness of the paper is that it only concentrates on a given application area, while it could have been interesting to consider the ideas and concepts in the paper in a more general framework.

**Questions:**

I do not have any specific point or question to the authors.
Possibly, if they could try to discuss their approach in a more general context (i.e., envisaging what could be done for other application areas) at various stages of the paper, it could make it stronger.

---

> ### Author Response · Authors · 2024-11-22
>
> Thank you for your time and feedback! We are glad that you enjoyed reading our work! We agree that we need alternative constructs to existing interpretability methods that focus on specific use cases.
>
> > In my view the only weakness of the paper is that it only concentrates on a given application area, while it could have been interesting to consider the ideas and concepts in the paper in a more general framework.
>
> We agree!
>
> We focused on consumer finance as a driving application area since: (1) there are rules and regulations that have been in place for over 30 years; (2) most companies are using feature attribution methods to comply with these laws. In this case, part of our motivation was that we could point to our results to highlight the need to revisit how we enforce and develop these laws in other areas.
> In general, we agree that we can use responsiveness scores in other tasks and domains. Specific application areas include:
> Healthcare: We can use responsiveness scores to evaluate decision models in health care (i.e. organ transplant, triage). The goal would be to increase transparency in these models and evaluate fairness (see [this article](https://www.ft.com/content/5125c83a-b82b-40c5-8b35-99579e087951) for an example of why a model could be unfair).
>
> Criminal Justice: In a similar fashion, we can evaluate the fairness of risk scoring models ([example](https://anonymous.4open.science/r/responsiveness-iclr-2025/figures/OGS.pdf)). Models will use features including criminal history and immutable attributes like age and sex. Although features like criminal history are not actionable, we can check their responsiveness to assess the model’s fairness. Simply put, we would expect a “clearing out” of one’s criminal history would lower their risk.
>
> Specific tasks include:
>
> - *Stress Testing*: For example, model developers can check if their model is robust to single-feature changes. In tasks like bot or spam detectors developers may want a model that is unresponsive to one-dimensional changes in a subset of features (i.e. they don’t want it to be too easy to manipulate).
> - *Eliciting Optional Features*: If a model allows end users to elicit optional features, we can use the responsiveness scores of the optional features to help users determine whether to provide them or not (see e.g., [Leemann et al. 2024](https://arxiv.org/abs/2210.13954)). This would lead to consent and transparency at prediction time.
>
> We updated the manuscript to discuss these tasks and areas. For now our plan is to introduce consumer finance as a motivating use case in Section 2, discuss extensions in Section 3 and discuss broader application areas in Related Work and Concluding Remarks. If you have any other ideas, we can account for them as well!

---

> > ### Comment · Reviewer_PaRQ · 2024-11-24
> >
> > Thanks for your reply, and your willingness to make changes to your paper.

---

### Official Review · Reviewer_hbQN · 2024-11-09

**Soundness:** 3
**Presentation:** 3
**Contribution:** 2
**Rating:** 6
**Confidence:** 4

**Summary:**

This paper addresses the limitations of feature attribution methods in decision-making scenarios where regulations mandate that individuals are informed of the principal reasons for adverse decisions. Existing methods often highlight important features without considering whether those features can actually be changed to achieve a desired outcome, thereby failing to provide actionable recourse.
The authors propose a new method for scoring feature responsiveness, which evaluates the likelihood that changing a particular feature can lead to a favorable model outcome.  The authors conducted an empirical study in consumer finance, demonstrating that their responsiveness-based approach more accurately identifies features that offer actionable recourse, flagging instances where standard methods provide “reasons without recourse.” They also release a Python library to support the implementation of feature responsiveness.

**Strengths:**

1. The paper effectively highlights a critical shortcoming in the use of feature attribution methods for generating explanations. By demonstrating that standard methods often provide non-actionable reasons, the authors reveal a key limitation that could undermine the intended goals of explainability and consumer protection in high-stakes applications.

2. The introduction of responsiveness scores offers a novel/nuanced approach to measuring "actionability." These scores help flag instances where individuals cannot achieve recourse, thereby preventing the issuance of misleading explanations and enhancing the practical value of model explanations.

**Weaknesses:**

(1) While I find the concept of feature responsiveness quite interesting, the contributions of this paper appear marginal, particularly in light of the work [1]. This paper draws upon existing ideas introduced in [1], such as the notions of reachable sets and action sets, which are used for recourse verification there (i.e., determining if an individual can achieve recourse through actions in the feature space). In [1], the approach returns a binary output: 1 if there exists an action that achieves recourse (feasible) and 0 if no such action exists (infeasible). In contrast, this work provides a probabilistic measure by assessing the fraction of actions on a feature that leads to recourse. Although the responsiveness score method offers a more nuanced probabilistic view, the framework in [1] could also be adapted to identify unresponsive features effectively. For example, the experiment of Table 3 (Recourse feasibility....) is similar to Table 2 in [1].

To improve the contributions, I would like to see this work go beyond identifying unresponsive features or "actionability" of features to leveraging the responsive score to produce better explanations. For example, can we leverage this responsiveness score to produce feature attribution methods that return more actionable features? Or when finding recourse (or counterfactual explanations) can we leverage the responsiveness score to find more actionable recourses?

(2) Generally, I also believe your method can go beyond checking the responsiveness of top-k features from a feature attribution method. You can also extend this to various recourse generation methods, counterfactual explanations, or any feature explainability methods.

(3) I challenge the assumption that feature attribution methods should necessarily provide actionable recourse. Attribution methods and recourse methods are fundamentally different types of explanations, each with distinct purposes and applications. While feature attribution methods are designed to identify and communicate the features that most strongly influence a model’s prediction, they are not inherently designed to satisfy an "actionability axiom". This distinction is important because blending the goals of feature attribution and recourse may lead to confusion and misaligned expectations. The paper might benefit from a brief discussion on this but I don't think the premise of "feature attribution method should be able to provide recourse" needs to be true.


(4) The definition of responsive score as a probability is a bit confusing as there is no random variable or distribution. It might be more precise to define the responsive score as a fraction or proportion. To accurately define responsive score as a probability you would need to properly characterize your sample space, probability measure, and events.

What if two features are individually irresponsive, but when considered together can be responsive (and can provide a recourse)?

(5) The responsiveness score does not consider the ease or difficulty of implementing certain actions, which is a crucial factor from the user's perspective. For instance, actions that are relatively easy to undertake, such as increasing a credit score by 2 points, should carry more weight than more challenging actions, increasing a credit score by 20 points. A weighted version of the responsiveness score, which factors in the feasibility or effort required for each action, could offer a more user-centric measure. This might better capture "user responsiveness" rather than a uniformly distributed action space.  For instance, what do you do when a feature is continuous and unbounded (e.g., income feature)?


minor --
Increase the text font size of Figure 1 for clarity.


[1] Kothari, Avni, et al. "Prediction without Preclusion: Recourse Verification with Reachable Sets." arXiv preprint arXiv:2308.12820 (2023).

**Questions:**

Address questions in the weakness section.

---

> ### Author Response · Authors · 2024-11-22
>
> Thank you for your time and feedback! We appreciate your comments and concerns, which have helped us to refine our paper. We would like to address them below:
>
> > I challenge the assumption that feature attribution methods should necessarily provide actionable recourse ... blending the goals...may lead to confusion and misaligned expectations....
>
> In short, *we agree!* There is no reason to expect feature attribution methods to provide recourse. This point is clear to us as well as to a growing body of researchers. As a recent example, see [Bilodeau et al. 2024](https://www.pnas.org/doi/abs/10.1073/pnas.2304406120) who show that feature attribution methods (i.e. SHAP) cannot reliably characterize model behavior.
>
> This point is *not* obvious to many people who matter – i.e.,  regulators, supervisors, and model deployers. To give you a sense of what things look like in the wild:
> - Consumer protection rules like the Adverse Action notice are designed – in part – to provide consumers with recourse (see e.g., [Barocas et al. 2020](https://dl.acm.org/doi/pdf/10.1145/3351095.3372830))
> - Industry watchdogs – i.e., the people that we would expect to highlight best practices – suggest that lenders comply with these rules using methods like LIME and SHAP (see e.g., this whitepaper by [FinRegLab](https://finreglab.org/wp-content/uploads/2023/12/FinRegLab_2023-07-13_Empirical-White-Paper_Explainability-and-Fairness_Insights-from-Consumer-Lending.pdf)
> - Regulators who enforce these laws are aware that different methods will highlight different solutions. However, they do not perceive these differences as "important enough" to provide specific guidance. In 2020, for example, the CFPB held a contest to how to build adverse action notices that were "actionable" [Adverse Action Sprint](https://www.consumerfinance.gov/rules-policy/competition-innovation/cfpb-tech-sprints/electronic-disclosures-tech-sprint/) and awarded the winning entry to a feature attribution method (SHAP).
>
> Overall, these points highlight a profound misunderstanding among stakeholders. These are people and organizations who care enough to write whitepapers and hold contests. They have the ability to improve how lenders comply with these rules – by changing their recommendations and requirements. However, they still support the use of feature attribution methods as explanations. Highlighting concerns with this usage is part of our goal in Section 5. We want to show that using feature attribution methods is not just "ineffective" (in that they highlight features that are unresponsive), but that it can also lead to harm (i.e., by providing reasons without recourse).
>
> > I would like to see this work go beyond identifying unresponsive features or "actionability" of features to leveraging the responsive score to produce better explanations.
> > I also believe your method can go beyond checking the responsiveness of top-k features from a feature attribution method.
>
> We were somewhat confused by these comments and wanted to flag them to make sure there wasn't a misunderstanding.
> The primary use case for responsiveness scores is to produce better explanations in tasks where we would like to provide recourse. In such tasks, we use feature responsiveness scores to highlight features directly; a resulting example explanation can be found on the right in Figure 1. We solely check the top-k features from another method as a means of highlighting the problem of reasons without recourse that currently exists with other methods; a key part of our paper is going beyond this to introduce feature responsiveness scores as an alternative explanatory method.
>
> Consider an application where we would provide a person with an explanation that contains up to 4 features. Here, a traditional explanation would highlight up to 4 features with the highest SHAP scores. In contrast, we would highlight up to 4 features with the highest responsiveness scores.
>
> We argue that our explanation is better because it highlights features that maximize the chances of recourse. It shows features with the highest proportion of single-feature actions that can change the prediction. More importantly, our **approach** can detect and mitigate cases where feature-highlighting explanations are misleading or harmful: (i) when individuals have no recourse; (ii) when individuals can only attain a desired prediction through joint intervention.
>
> Having said that, we agree that we can apply responsiveness scores to other tasks and applications. We list these in our response to [PaRQ](https://openreview.net/forum?id=wsWCVrH9dv&noteId=iMevY6xP8z) and will incorporate them in our paper.

---

> ### Author Response · Authors · 2024-11-22
>
> > This paper draws upon [...] ideas introduced in [1] [...] Although the responsiveness score method offers a more nuanced probabilistic view, the framework in [1] could also be adapted to identify unresponsive features effectively.
>
> We want to be clear that we do not just "draw" on these ideas in [1] but have developed them considerably to address a fundamentally different problem (how to highlight responsive features) and highlight a fundamentally different message (LIME and SHAP often provide reasons without recourse, demanding specialized solutions)
>
> We have three main contributions – all distinct from those in [1] and valuable to several audiences:
> 1. *Responsiveness Scores*: We present a different paradigm for feature attribution that highlights how these techniques can be used to do more than just score (e.g., flag "fixed predictions"). Responsiveness scores can also readily be used to improve consumer protection rules that affect millions in applications like lending and hiring. More broadly, we believe responsiveness scores have a broad range of applications. We discuss a few variants and extensions below and in our response to [PaRQ](https://openreview.net/forum?id=wsWCVrH9dv&noteId=iMevY6xP8z).
>
> 2. *Algorithms & Software*: We propose machinery to construct reachable sets for any model and any dataset in a way that scales and has formal guarantees. We construct reachable sets over continuous features via sampling (this is new), and over discrete feature spaces via enumeration (this extends the approach in [1]).  We see this as a significant contribution since we can now compute responsiveness scores for all possible use cases – i.e., for any model class, any dataset. If we were to simply apply [1], then we would be limited to discrete feature spaces and would suffer from scalability problems.
>
> 3. *Empirical Results*: Our study demonstrates how current practices to comply with consumer protection rules can backfire. We show that explaining predictions can mislead individuals by highlighting reasons without recourse. Our findings highlight the extent of this problem across datasets, model classes, and under conservative assumptions. They reflect the kind of evidence that lawmakers need to revisit rules and enforcement.
>
>
> > A weighted version of the responsiveness score... could offer a more user-centric measure.
>
> Thanks for bringing this up! We agree! We've now included the following examples in the manuscript:
>
> - Robust Version: We consider whether our recourse guarantee for single-feature action is sensitive to changes in other features. If we are listing features for recourse, we might want them to be robust against changes in other features. This is because consumers may (inadvertently) act upon other features.
> - Cost-Weighted Version: We agree that the responsiveness score could be changed to account for the ease or difficulty of actions and can include this version of the responsiveness score in Section 3. To give some background, this "weighted" variant is easy to implement: given a cost function, we can compute the cost of each reachable point and return a feature’s cost-weighted responsiveness (some variant of multiplying the cost by responsiveness of the action). We made a deliberate decision to not include the weighted version as a "default" version because it may invalidate the natural probabilistic interpretation. In practice, it is challenging to specify a cost function: costs will vary from person to person and should be elicited from individuals.
>
> Part of the reason why we compute our scores using reachable sets is because they make it easy to implement variants of the responsiveness score. In this case, all variants can be implemented in only a few lines of Python code – without compromising versatility nor scalability as they involve basic matrix-vector operations. We're happy to include short demos in the Appendix or notebooks in the code if you'd like.

---

> ### Author Response · Authors · 2024-11-22
>
> > Can we leverage.. to produce feature attribution methods that return more actionable features?
>
> We can!
>
> In addition to our introduced feature responsiveness scores - which are directly a new feature attribution method with actionable features - one way to leverage responsiveness scores is to use them as an "actionability filter." In this approach, we would only include a feature in an explanation if it exceeds a desired degree of responsiveness – i.e., if $\mu_j(\mathbf{x}) ≥ \tau$. In our experiments, we report results for “action-aware" versions of LIME and SHAP, which can be viewed as the simplest implementation of this strategy.  Specifically, these methods would output a feature-highlighting explanations that includes the features with the top-4 scores from LIME and SHAP, choosing only features that are *actionable* (i.e., for which $\|A_j(\mathbf{x})\| \geq 0$).
>
> Our results show how this strategy can improve the responsiveness of feature-highlighting explanations by filtering out immutable features. However, it may still lead to cases where individuals could fail to achieve recourse because: (i) changing a feature induces changes in other features; (ii) changing a feature may not flip a prediction. In practice, we could resolve these issues by reporting features that achieve a threshold degree of responsiveness. In this case, however, the resulting explanation may not maximize the chances of recourse (e.g., because the features might have low responsiveness). Both of these concerns are resolved by using feature responsiveness scores.
>
> > Can we leverage this responsiveness score... to find more actionable recourses?
>
> Yes we can! One way is to use them as filters for counterfactual explanations as described above. More broadly, however, we can use the reachable sets produced by our method to characterize important properties of recourse. For example, we can use the reachable set to determine if a feature is **monotonically** responsive; that is, whether we can attain a desired prediction through monotonic actions and preserve the target prediction with actions of larger magnitude. This property may be valuable because it informs consumers whether they need to worry about taking actions that are “too big.” Notationally, this would equate to checking if $h(\mathbf{x}_0 + \mathbf{m}) = 1$ for all $\mathbf{m} \in \{\mathbf{a} | \mathbf{a}_j > t\}$ where $t \in \mathbb{R}$ is some threshold value.
>
> > What if two features are individually irresponsive, but when considered together can be responsive (and can provide a recourse)?
>
> It is definitely possible that there may be a set of features that are not individually responsive but are collectively responsive.
> Consider a simple classifier like: $h(x_1,x_2) =\mathbb{1}[x_1 + x_2 + 2x_1x_2 \geq 3]$ where the individual point of interest is $\mathbf{x}_0 = (0, 0)$. Clearly, features $x_1$ and $x_2$, individually, are not responsive. This doesn’t mean that they have a fixed prediction: when we act on both of them, the prediction changes.
>
> You can see this in Table 3 where we report the proportion of people who *only* have recourse through actions that change 2+ features (*% nD* metric). We could account for such cases in our responsiveness score. Part of the reason why we do not is because of the current regulatory landscape as well as concerns about user understandability. It is difficult to portray multi-feature changes like the one above because feature-highlighting explanations like adverse action notices are lists of features that do not characterize feature dependencies or interactions. Simply put, consumers who see them are likely to believe that they can be changed independently.
>
> > The definition of responsive score as a probability is a bit confusing [...] It might be more precise to define the responsive score as a fraction or proportion...
>
> Thanks for this! We agree and prefer this interpretation as well! We've updated the manuscript to use this interpretation throughout. We have kept probability notation in Definition 4 but can change it to an expectation if you'd like.
>
> > minor -- Increase the text font size of Figure 1 for clarity.
>
> Fixed!

---

> > ### Author Response · Authors · 2024-11-26
> > **Checking In**
> >
> > We just wanted to check in and see if you’ve had a chance to read our comments! We have a few more days left in the discussion period, so if you have additional questions or concerns, let us know, and we can do our best to address them over the coming days. Alternatively, if we have addressed your concerns, please consider raising your score!

---

> > > ### Comment · Reviewer_hbQN · 2024-12-01
> > >
> > > Thank you for your detailed rebuttal and making changes to the paper, particularly clarifying your contributions with respect to [1].
> > > I have raised my score.

---

### Author Response · Authors · 2024-11-22

We thank all reviewers for their time and feedback!

We were thrilled to see that all reviewers recognized the positive aspects of our paper – i.e., that it studies a problem that is  “interesting” and “very important” [6js1] and highlights a “critical shortcoming” of existing methods [hbQN] through a “comprehensive” empirical study [pPyj]. We were also glad to see that reviewers recognized our proposed solution as “novel” [hbQN, pPyj], “rigorous” [PaRQ], and with “a lot of potential use cases and extensions” [PaRQ].

Overall, our reviews revealed several areas where our manuscript would benefit from additional discussion or clarification (e.g., motivating consumer finance as a driving application, describing a list of other tasks and applications, and including a figure to highlight the conceptual difference with traditional recourse methods). We have already addressed some of these in our revision (e.g., adding missing content, improving figures, fixing typos), and will address remaining concerns based on the outcome of our discussion. We look forward to engaging with everyone over the coming days.

---

### Meta-Review · Area_Chair_2wC7 · 2024-12-22

**Metareview:**

This paper sits broadly in the XAI space, more specifically the recourse portion of XAI - recourse roughly meaning a set of steps an input might take to adjust its features such that it crosses a decision boundary from a negative outcome to a positive one.  Reviewers appreciated the focus of the paper - there's a lot of contention around XAI writ large, but recourse (in part by way of adverse action notices and their analogues) is one of the areas that is unquestionably addressing a real-world use case.  Doubling down on this, reviewers appreciated the critical take on feature attribution methods, on actionable features, and a slew of other real-world concerns.

**Additional Comments On Reviewer Discussion:**

We appreciate the authors' in-depth rebuttals.  Reviewer engagement was light, but this AC believes the authors addressed many of the reviewers' concerned, and will appreciate an updated future version.

---

### Decision · Program_Chairs · 2025-01-22

Accept (Poster)